# A three-quantile bias correction with spatial transfer for the correction of simulated European river runoff to force ocean models

Stefan Hagemann[*], Thao T. Nguyen and Ha T. M. Ho-Hagemann

Institute of Coastal Systems – Analysis and Modelling, Helmholtz-Zentrum Hereon, Max-Planck-Str. 1, 21502 Geesthacht, Germany

[*] Correspondence: Dr. Stefan Hagemann, stefan.hagemann@hereon.de

**Abstract**
In ocean or Earth system model applications, the riverine freshwater inflow is an important flux
affecting salinity and marine stratification in coastal areas. However, in climate change studies,
the river runoff based on climate model output often has large biases on local, regional or even
basin-wide scales. If these biases are too large, the ocean model forced by the runoff will drift
into a different climate state compared to the observed state, which is particularly relevant for
semi-enclosed seas such as the Baltic Sea. To achieve low biases in riverine freshwater inflow
in large-scale climate applications, a bias correction is required that can be applied in periods
where runoff observations are not available and that allows spatial transferability of its
correction factors. In order to meet these requirements, we have developed a three-quantile bias
correction that includes different correction factors for low, medium and high percentile ranges
of river runoff over Europe. Here, we present an experimental setup using the Hydrological
Discharge (HD) model and its high-resolution (1/12°) grid. First, bias correction factors are
derived at the locations of the downstream stations with available daily discharge observations
for many European rivers. These factors are then transferred to the respective river mouths and
mapped to neighbouring grid boxes belonging to ungauged catchments. The results show that
the bias correction generally leads to an improved representation of river runoff. Especially
over Northern Europe, where many rivers are regulated, the three-quantile bias correction
provides an advantage compared to a bias correction that only corrects the mean bias of the
river runoff. Evaluating two NEMO ocean model simulations in the German Bight indicated
that the use of the bias corrected discharges as forcing leads to an improved simulation of sea
surface salinity in coastal areas. Although in the present study, the bias correction is tailored to
the high-resolution HD model grid over Europe, the methodology is suitable for any high-
resolution model region with a sufficiently high coverage of river runoff observations. It is also
noted that the methodology is applicable to river runoff based on climate hindcasts as well as
on historical climate simulations where the sequence of weather events does not match the
actual observed history. Therefore, it may also be applied in climate change simulations.
**Keywords:** Bias correction, river runoff, discharge, high resolution, Europe, sea-surface
salinity

# 1  Introduction

River runoff (or discharge/streamflow) is an important component of the global hydrological cycle, accounting for about one-third of precipitation over land areas. It closes the water cycle between land and ocean and influences various ocean properties, in particular the salinity of coastal and semi-enclosed seas (e.g. Väli et al., 2013), the ocean stratification in shelf areas (e.g. Hordoir and Meier, 2010) such as the German Bight (Becker et al., 1992), and the thermohaline circulation in different regions (e.g. Hordoir et al., 2008; Lehmann and Hinrichsen, 2000; Marzeion et al., 2007). In addition, river runoff and associated nutrient loads are important factors influencing marine ecosystem functioning (Daewel and Schrum, 2017).

Consequently, river runoff needs to be adequately represented in studies of the impacts of climate change on the marine environment or in coupled Earth system studies. In such studies, the atmospheric data used to force the respective ocean model are usually taken from climate models, reanalysis products or hydrological models. Here, it is desirable that the river runoff is consistent with the atmospheric forcing (e.g. Vinayachandran et al., 2015; Hagemann and Stacke, 2022), i.e. that the impact of weather events and trends in the atmospheric forcing is transferred via the river runoff into the ocean. In previous modelling studies, runoff was often taken from climatology or discharge observations, especially when hindcasts were used. However, this is not a recommended approach for climate change studies where consistently simulated river runoff should be used. Runoff from the driving climate, land surface or hydrological model will contain biases, e.g., due to biases in precipitation and/or uncertainties in the land surface representation of the model. Many simulations of historical daily river runoff show common biases in the tails of their distributions, with high discharges underestimated and low discharges overestimated (Farmer et al., 2018, and references therein). If the basin-wide biases are too large, a bias correction of the simulated discharge would be necessary to avoid the ocean model drifting into a different climate state compared to the observed state. This is particularly relevant for semi-enclosed seas such as the Baltic Sea. For example, for Baltic Sea ocean models, the mean long-term bias of river runoff must be less than 7% (Hagemann and Stacke, 2022).

The bias correction of river runoff is an approach that has been used particularly for short-term hydrological forecasts and ensemble predictions of up to six months. However, these approaches (see, e.g., those listed in Kim et al., 2021; Madadgar et al., 2014)are often specifically trimmed to flood forecasts. Therefore, they often require the existence of observed values from previous time steps so that that are not applicable in climate change studies, such as autoregression models (Kim et al., 2021) or components of a Bayesian forecasting system (Krzysztofowicz and Maranzano, 2004). Others like non-parametric methods based on Bayesian approaches as proposed by Brown and Seo (2010; 2012) need a large number of ensemble members (Madadgar et al., 2014).

Recently, bias correction of river runoff has also been applied in the context of climate change. Quantile mapping based approaches are often used for such bias correction, as this usually leads to a large improvement in the representation of discharge of the considered river. For example, Budhathoki et al. (2022) used quantile mapping to correct discharge bias in the Chao Phraya River basin (Thailand), and Daraio (2020) used it for two basins in New Jersey (USA). A criticism of using quantile-mapping in flood forecasting is that it does not maintain the pairing of corresponding simulated and observed flows (Madadgar et al., 2014). Madadgar et al. (2014) also noted that quantile mapping was not always successful in improving the initial forecast trajectory. In their application for the Sprague River (southern Oregon, USA), the skill

of the forecast actually deteriorated when the quantile mapping technique was used. Similarly,
Malek et al. (2022) used a quantile mapping based bias correction of discharge and showed that
ex-post corrections of simulated discharge do not necessarily reduce biases in the simulation of
key processes and in some cases can severely degrade system simulations.
Consequently, the aim of the present study was to develop a bias correction method sufficient
to meet the requirements of ocean models in large-scale climate change studies. Note that we
did not aim for the most accurate reproduction of observed discharge characteristics, as required
for short-term hydrological predictions and flood forecasts used by water resource decision
makers (e.g. Shi et al., 2008). In order to maintain a high degree of temporal consistency of
simulated runoff with the meteorological patterns in the driving (on- or offline) climate model
(or data), a bias correction with as little fitting or modification of the daily sequence of runoff
curves as possible is desired. Thus, our target is a simple bias correction that corrects the mean
bias and the tail biases of the discharge distribution in climate change applications of ocean or
coupled system models. The bias correction factors should be transferable from downstream
stations to river mouths as well as to neighbouring ungauged catchments. Furthermore, it should
be applicable to climate model or Earth system model data that lack the observed sequence of
actual discharge events. Therefore, we decided to not apply methods that employ detailed
modifications of the discharge curves for specific rivers such as those methods that use complex
matrix arithmetic of observed and simulated discharge time series (e.g. Zhao et al., 2011), or
the common quantile-mapping approaches, The latter are conducted using a lot of bins, so that
the bias in the discharge curve of a specific river can be strongly reduced. However, these
detailed correction factors for every bin may likely not be transferred to other locations. It may
work for the same river if station and river mouth are relatively close to each other, but certainly
may not be valid for the transfer to neighbouring catchments.
The manuscript is organised as follows. Section 2 describes how the simulated discharges
were generated and the newly developed bias correction methodology, as well as the data,
models and metrics used in this study. Sections 3 and 4 evaluate the simulated and bias corrected
discharges and present the effects of the bias correction for station locations and sea basin
inflows, respectively. Finally, Section 5 concludes with a summary and conclusions.

## 2   Data and Methods

To generate the freshwater inflow from rivers to the ocean, we used an experimental setup
analogous to Hagemann and Stacke (2022). Here we used two atmospheric forcing datasets
(Sect. 2.1) and the same model chain of two large-scale hydrological models. The global
hydrological model HydroPy (Sect. 2.2) was used to generate the input to the Hydrological
Discharge (HD) model (Sect. 2.3) at the resolution of the atmospheric forcing data ($0.5°$). These
input data of surface and sub-surface runoff were then interpolated onto the HD model grid and
the HD model was used to simulate daily discharges from land to sea. Subsequently, we bias
corrected these time series as described in Section 2.4 to generate bias corrected discharges at
coastal ocean boxes of the European HD model domain from 1901-2019. Note that we
combined the simulations based on two different atmospheric forcing datasets to cover the
whole $20^{th}$ century and to include the more recent years in the bias corrected discharge time
series. Such an approach was also used in the second phase (ISIMIP, 2023) of the Inter-Sectoral
Impact Model Inter-Comparison Project (ISIMIP; Warszawski et al., 2014). Figure 1
summarises the experimental setup. Section 2.5 refers to the observational data that are used in
the evaluation of the model results. Finally, the evaluation metrics used in the analysis of the
results are presented in Sect. 2.7.

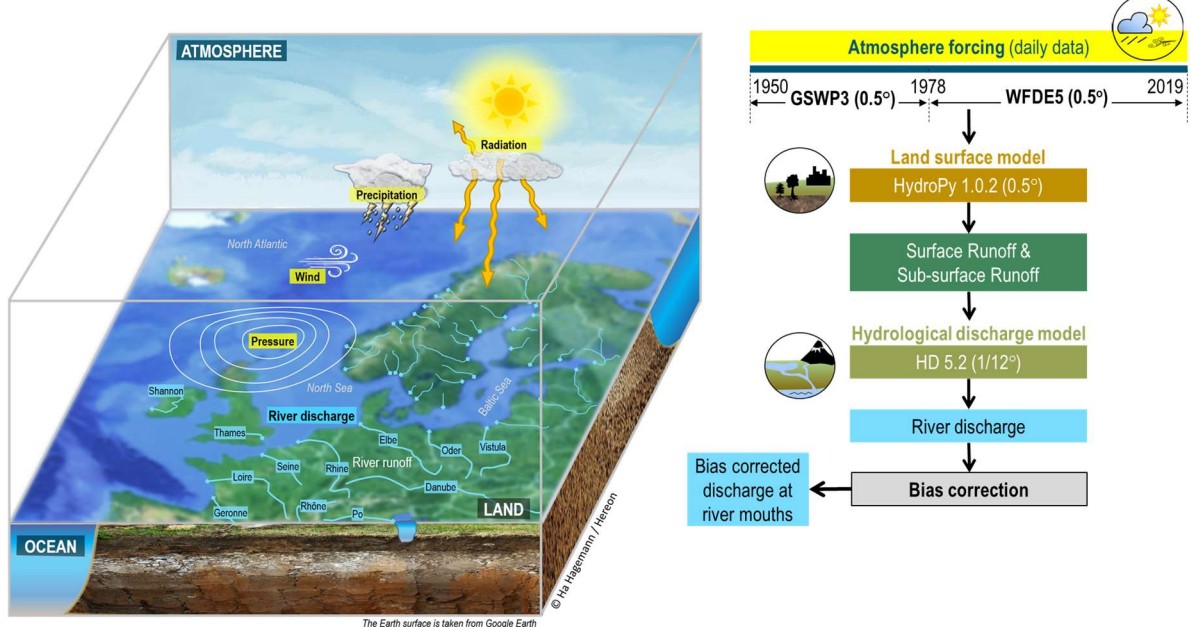



**Figure 1.** Overview on the main steps of generating bias corrected river discharge at HD
river mouths.

## 2.1 Atmospheric forcing

We used two atmospheric datasets comprising daily data of various near-surface atmospheric
variables. They have been used as meteorological forcing datasets in several climate impact
assessments and are recommended by ISIMIP (2023). Both datasets were specifically generated
to force global hydrological models for hindcast simulations. They are based on re-analysis
products from different weather forecast centres and bias-correction procedures were applied
by the respective creators to improve their data.
The Global Soil Wetness Project Phase 3 (GSWP3; Dirmeyer et al., 2006; Kim, 2017)
dataset is available at 0.5° resolution from 1901-2014. To generate the GSWP3 dataset, Kim
(2017) dynamically downscaled the 20[th] Century Reanalysis (Compo et al., 2011) onto the T248
(~0.5°) grid using a spectral nudging technique (Yoshimura and Kanamitsu, 2008) in a Global
Spectral Model. Observation-based bias correction procedures were then applied to the
downscaled data to obtain daily time series.
To generate the WFDE5 dataset, Cucchi et al. (2020) applied the WATCH Forcing Data
methodology (Weedon et al., 2011) to surface meteorological variables from the ERA5
reanalysis (Hersbach et al., 2020) to obtain bias corrected time series. ERA5 is the fifth
generation of atmospheric reanalysis produced by the European Centre for Medium-Range
Weather Forecasts (ECMWF). WFDE5 is provided at 0.5° spatial resolution from 1979-2019.
Mengel et al. (2021) stated that WFDE5 is considered as the more realistic dataset, especially
with respect to day-to-day variability for variables for which the monthly mean values were
bias corrected, such as precipitation and temperature. For more information on application and
evaluation of both datasets, see, e.g., Mengel et al. (2021) and references therein, Hassler and
Lauer (2021), (Arora et al., 2023).

## 2.2 HydroPy setup

HydroPy (Stacke and Hagemann, 2021) is a state-of-the-art global hydrology model for
which no model calibration was performed for its setup. Within global hydrological modelling,
the usage of uncalibrated models is rather common (see, e.g., Haddeland et al., 2011), even
though some models exist that are calibrated for global studies. In the present study, HydroPy
was driven by daily forcing data from 1901-2019. Daily input fields of surface and subsurface
runoff were generated at a resolution of 0.5°. Analogous to the ERA5 forced simulation in
Hagemann and Stacke (2022), precipitation, 2m temperature, downwelling shortwave and
longwave radiation, 2m specific humidity, surface pressure and 10m wind are used as forcing
from the respective forcing dataset. We performed a spin-up simulation over 50 iterations of
the year 1901 with the GSWP3 forcing (cf. Stacke and Hagemann, 2021) to initialize the
storages in the HydroPy model and to avoid any drift during the actual simulation period. We
then forced HydroPy with the GSWP3 data from 1901-1978 and continued with the WFDE5
data from 1979-2019. We also conducted a GSWP3 forced simulation from 1979-2014 in order
to derive bias correction parameters for the earlier period. For our analysis, we focus on the
years from 1950 onwards so that we have an additional transient spin-up of 49 years.

## 2.3 HD model setup

The HD model (Hagemann et al., 2020) is a well-established river routing model that is
implemented in a range of global and regional model systems. As noted in Hagemann et al.
(2020), no river specific parameter adjustments were conducted in the HD model to enable its
applicability for climate change studies and over catchments, where no daily discharges are
available at a downstream station. To simulate discharge with the HD model, we used the daily
input fields of surface and subsurface runoff that were generated by HydroPy from the GSWP3
and WFDE5 data (see Sect. 2.2). As the time step of these runoff data is one day, the time step
of the HD model was also set to one day. However, an internal time step of 0.5 hours is used
for the flow within the river, as the minimum travel time through a grid box is limited by the
chosen time step. The HD model v5.2.0 (Hagemann et al., 2023) was applied over the European
domain, which covers the land areas between -11°W to 69°E and 27°N to 72°N. The domain,
along with a number of rivers specifically noted in this study, is shown in Figure 2. In the
following, we refer to the WFDE5-based discharges as HDW and to the GSWP3-based
discharges as HDG. The corresponding bias-corrected discharges are referred to as HD-BC in
general and HDW-BC and HDG-BC in particular.

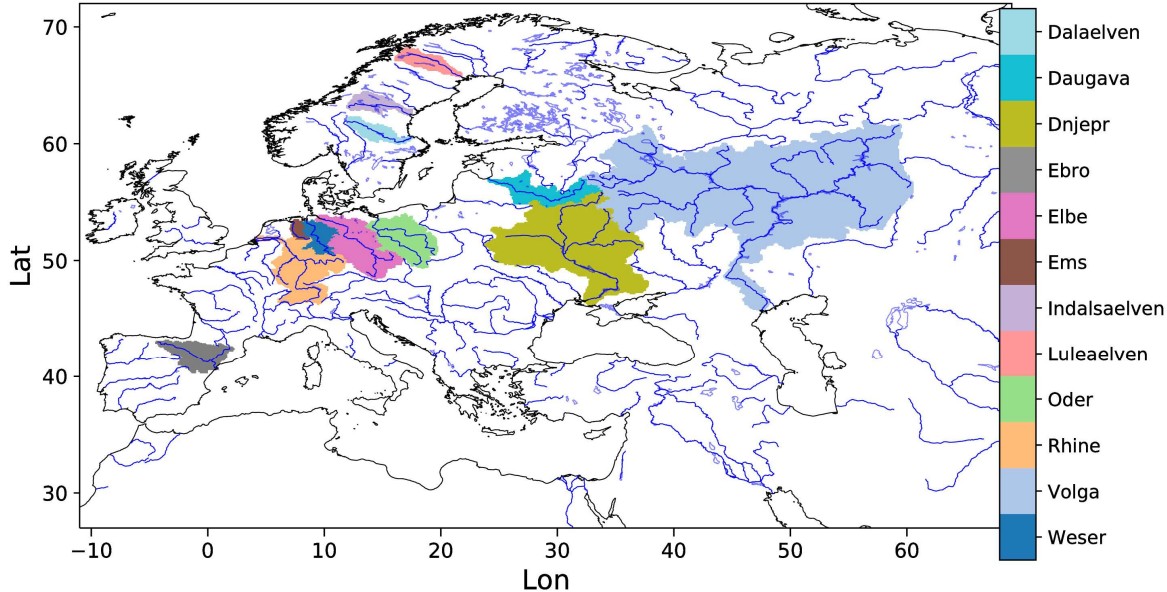

**Figure 2.** European HD model domain and catchment areas for selected rivers

## 2.4 Bias correction of river runoff

We have developed a bias correction method for river runoff that uses correction factors for three quantiles and includes a spatial transfer of these factors. We note that our three-quantile bias correction is similar to a very coarse quantile mapping. The latter has been introduced in climate change impact research to correct for significant biases in data produced by global and regional climate models. Quantile mapping is a distribution mapping in which the distribution function of climate values is corrected to match the observed distribution function. Details of such mapping applied to precipitation and surface air temperature can be found, for example, Piani et al. (2010) and Teutschbein and Seibert (2012). Our bias correction method involves several steps. First, different correction factors for low, medium and high percentiles are calculated at the station locations and then applied at the respective river mouths. Finally, an interpolation is performed to neighbouring coastal mouth points for which no downstream observations are available in the respective catchment. This procedure is summarised in Figure 3. The three percentile ranges for daily discharge $q_i$ are classified by

- Low (L): $q_i \leq Q_p$
- Medium (M): $Q_p < q_i < Q_{100-p}$
- High (H): $q_i \geq Q_{100-p}$

Here, $Q_p$ denotes the $p^{th}$ percentile of the daily discharge and $p$ was set to 20. The percentiles $Q_p$ and $Q_{100-p}$ were determined separately for the observed and the simulated discharges at the downstream station locations and then the mean discharges $\overline{q_R}$ were calculated for the three percentile ranges $R \in \{L, M, H\}$. Note that for these calculations only those days were considered for which an observed discharge was available. Then, the mean bias $b_R$ (in %) was calculated for each percentile range and a correction factor $f_R$ to remove the bias was derived as

$$f_R = \frac{100}{b_R + 100}$$

210

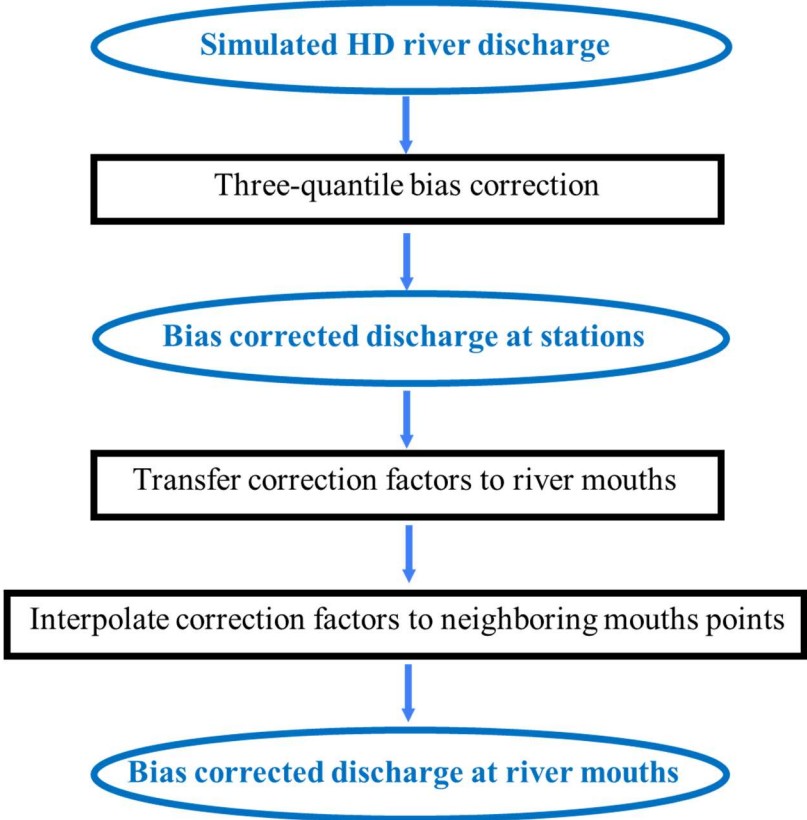

211

Figure 3. Steps to derive bias corrected discharge at river mouths from simulated
discharges.

For the evaluation of the bias correction in Sect. 3, these correction factors were applied to
the simulated discharges at the station locations. As the correction factors are independent of
the absolute amount of discharge, they could be applied to the respective river mouths. For each
river mouth with more than one inflow ($j > 1$) for which a correction factor $f_{R,j}$ is determined, a
combined correction factor is obtained by calculating an average weighted by the respective
mean inflows $Q_j$.

$$\overline{f_R} = \frac{\sum_j f_{R,j} * Q_j}{\sum_j Q_j}$$

From these river mouths, an interpolation is performed to neighbouring coastal mouth points
for which no downstream observations are available in the respective catchment. This
interpolation was motivated by the fact that the general pattern of bias of neighbouring rivers
is often similar (cf. Sect. 3.1). The interpolation is performed by inverse distance weighting
from the four closest (or fewer) river mouths within a search radius of 200 km. If no river mouth
with a correction factor was found within the search radius, the correction factor was set to one
(i.e. no correction).

Note that the bias correction can lead to spurious daily jumps in discharge when the
percentile boundary is crossed and the bias correction factors differ between the percentile

ranges. In order to reduce this effect, a smoothing radius of $\Delta s = 0.05$ was introduced around
the percentile boundaries, which was applied at both station locations and river mouths.
For $(1 - \Delta s) * Q_p < q_i < (1 + \Delta s) * Q_p$:
$$\tilde{q}_i = q_i * \left( f_L + (f_M - f_L) * \frac{(q_i - (1 - \Delta s) * Q_p)}{2 * \Delta s * Q_p} \right)$$
For $(1 - \Delta s) * Q_{100-p} < q_i < (1 + \Delta s) * Q_{100-p}$:
$$\tilde{q}_i = q_i * \left( f_M + (f_H - f_M) * \frac{(q_i - (1 - \Delta s) * Q_{100-p})}{2 * \Delta s * Q_{100-p}} \right)$$
The bias correction procedure corrects the days that fall into the different percentile ranges.
However, this does not necessarily mean that it also corrects the whole distribution into the
three percentile ranges. Particularly, if the biases in these ranges are quite different, the days
may change their class and order within the distribution.
In order to apply the three-quantile bias correction to the simulated discharge time series
from 1901-2019, two sets of bias correction factors were derived. The first set uses HDW and
discharge station observations for the period 1979-2014. This set was used to bias correct the
simulated discharge at HD river mouths from 1979-2019. The second set uses a further
discharge simulation where we continued HDG utilizing the GSWP3 forcing up to 2014. Again,
the set of bias correction factors was derived for the period 1979-2014 using discharge station
observations. This set was then used to bias correct the simulated discharge at the HD river
mouths from 1901-1978.

## 2.5 Observed discharge data

We used available daily discharge data from downstream gauges for many rivers across Europe
with a catchment area of about 1000 km² or more. These station data were obtained from Global
Runoff Data Centre  and various agencies and institutions listed in table 2 of Hagemann and
Stacke (2022). In addition, French discharge data were accessed from the E.U. Copernicus
Marine Service Information. In order to allow an assessment of the discharge at the river
mouths, we considered basin-wide estimates from three different sources.
For the Baltic Marine Environment Protection Commission – also known as the Helsinki
Commission (HELCOM), Svendsen and Gustafsson (2022) provided annual waterborne
inflows into the seven main sub-basins of the Baltic Sea (Figure 4 – upper panel) from 1995 to
2020. Waterborne inflows comprise the sum of river runoff and direct inflows, i.e. flows from
point sources discharging directly into the Baltic Sea. These point sources are not included in
our experimental setup or in the bias correction. However, their contribution to the total
waterborne inflow to the Baltic Sea is only about 1% (HELCOM, 1998).

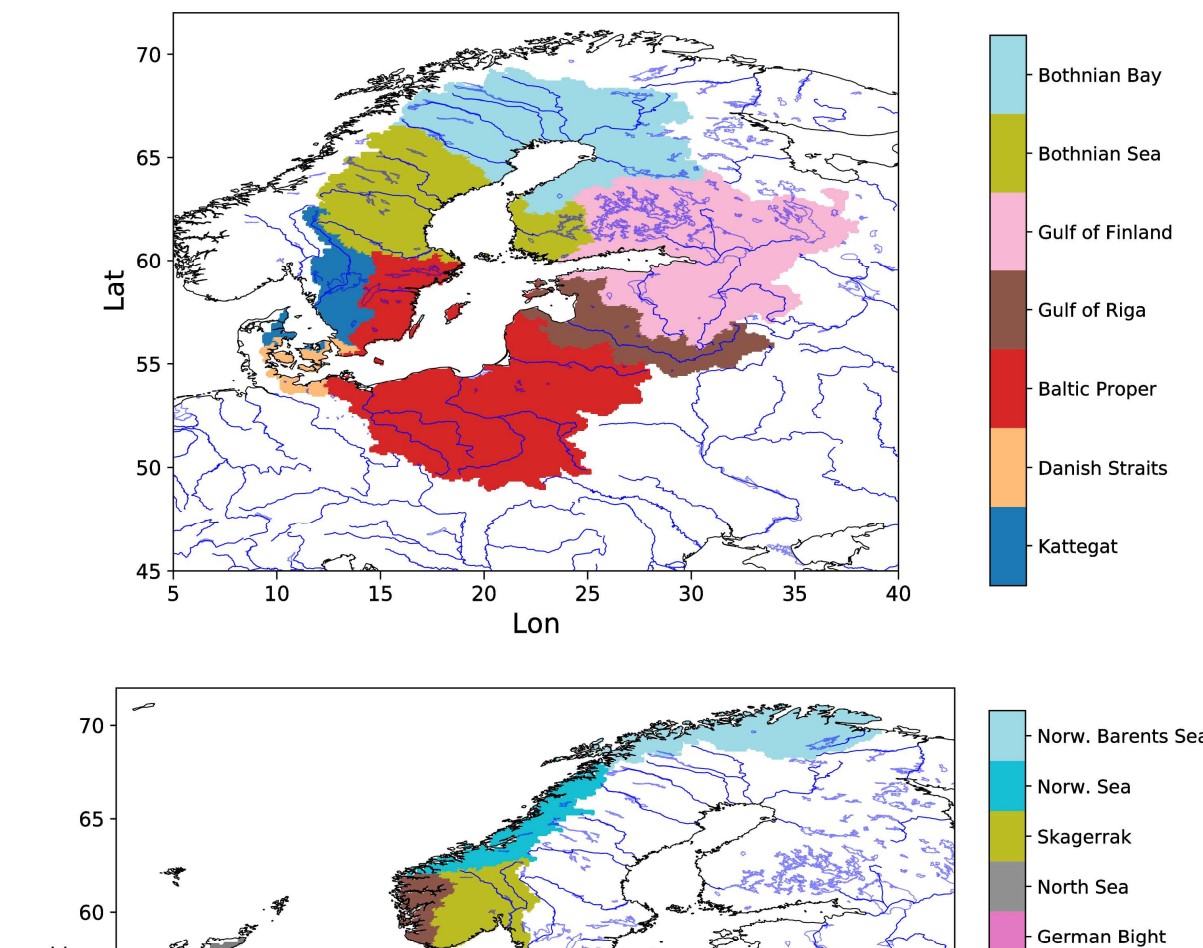

**Figure 4.** Selected HELCOM (upper panel) and OSPAR (lower panel) basins for which inflows are considered. For OSPAR, the Spanish Atlantic basin is limited to the coast of Northern Spain.

Under the umbrella of the OSPAR Convention (Convention for the Protection of the Marine Environment of the North-East Atlantic), the IGC-EMO (Intersessional Correspondence Group for Eutrophication Modelling) database (Lenhart et al., 2010) of daily riverine freshwater inflows and nutrient loads was compiled by Van Leeuwen and Lenhart (2021), covering the major rivers discharging into the Baltic Sea, the North Sea and the Northeast Atlantic. An updated database covering a total of 370 rivers was mapped onto the flow grid of the European 1/12° domain of the HD model by Van Leeuwen and Hagemann (2023). The associated catchment areas of these rivers, which flow into a particular specific sea basin, do not cover the entire catchment area of the respective basin (see Table 1) so that the total inflow of the sea

basin is underrepresented by the IGC-EMO data. To generate basin-wide estimates, we have up-scaled these values by dividing the integrated IGC-EMO river discharges in a basin by the fractional coverage of the entire basin catchment on the HD grid. Basin estimates for which the fractional coverage is less than 75% are considered to be highly uncertain and are therefore provided for completeness only, but are not included in the assessment of simulated inflows.

**Table 1.** Sea basin catchment areas on the HD model grid and the fractional catchment coverage of the associated IGC-EMO rivers.

| | HD Area [km²] | | |
|---|---|---|---|
| Sea basin | IGC-EMO | Total | Coverage |
| Baltic Sea | 1513967 | 1671823 | 90.6% |
| Bothnian Bay | 238898 | 258420 | 92.4% |
| Bothnian Sea | 199908 | 219375 | 91.1% |
| Gulf of Finland | 379628 | 412412 | 92.1% |
| Gulf of Riga | 124386 | 134025 | 92.8% |
| Baltic Proper | 494929 | 551295 | 89.8% |
| Danish Straits | 6731 | 19417 | *34.7%* |
| Kattegat | 69487 | 76876 | 90.4% |
| Norwegian Barents Sea | 0 | 81004 | 0.0% |
| Norwegian Sea | 0 | 58408 | 0.0% |
| Skagerrak | 89060 | 101787 | 87.5% |
| North Sea | 514334 | 599755 | 85.8% |
| German Bight | 201233 | 208807 | 96.4% |
| Norwegian North Sea | 4590 | 31327 | *14.7%* |
| English Channel | 94327 | 122235 | 77.2% |
| Celtic Sea | 41122 | 44845 | 91.7% |
| Irish Sea | 29748 | 35584 | 83.6% |
| French Atlantic | 207657 | 257981 | 80.5% |
| Northern Spanish Atlantic | 17692 | 46574 | *38.0%* |

In addition, we used estimates of long-term mean sub-basin-wide inflows to the North Sea and Northeast Atlantic, published directly by OSPAR (Farkas and Skarbøvik, 2021). Figure 4 (lower panel) shows the selected OSPAR basins for which the inflows are considered. It should be noted that the sea basin inflows provided by the different OSPAR countries are not consistent. Some countries include discharge estimates for unmonitored areas, while others do not (Table 2). In addition, the sea basin catchment coverage of the monitored areas varies between the countries. Note also that we have excluded the Spanish Atlantic from our comparisons for the following reason. Here, we limited the Spanish Atlantic basin to the coast of northern Spain (see Figure 4 – lower panel) to allow a comparison with the IGC-EMO data as the IGC-EMO data only cover rivers in this region, hereafter referred to as NSpA. These rivers cover about 38% of the total NSpA area on the HD model grid (Table 1), while the OSPAR data for NSpA cover about 50% (23201 km²; Farkas and Skarbøvik, 2021). However, the associated IGC-EMO discharge from 1961-1990 (629 m³/s) is 75 % larger than the OSPAR long-term mean average (359 m³/s). Therefore, both inflow values are unlikely to be representative for the NSpA region and this region is not considered in the following.

**Table 2.** Country catchment coverage of OSPAR data and inclusion of estimates for
unmonitored areas (Borgvang et al., 2008). NI means that no information on the
coverage was provided.

| Country | Coverage | Unmonitored |
|---|---|---|
| Belgium | > 90% | No |
| Denmark | NI | Yes[3] |
| France | 84% | Yes |
| Germany | >90% | No[1] |
| Ireland | NI | Yes |
| Netherlands | >90% | No |
| Norway | ca. 50% | Yes |
| Portugal | NI | No |
| Spain | NI | No |
| Sweden | 88.7% | Yes |
| United Kingdom | ca. 80%[2] | No |

[1] Only for Eider river
[2] 10% in direct discharge
[3] e.g. Farkas and Skarbøvik (2021)

## 2.6 Ocean model experiments

To assess the effect of using bias corrected river discharge on simulated salinity in the German
Bight, we used version 3.6 of the Nucleus for European Modelling of the Ocean (NEMO;
Madec et al., 2017) and adopted a domain setup used by Ho-Hagemann et al. (2020). This
domain covers the region of the north-west European shelf, the North Sea and the Baltic Sea
between 19.89 E to 30.16 E and 40.07 N to 65.93 N with a resolution of two nautical miles (ca
3.6 km). We used the atmospheric forcing from ERA5 and the ocean boundary forcing from
the ECMWF Ocean Reanalysis System 5 (ORAS5; Zuo et al., 2019) to conduct two simulations
from 2010 to 2018. Initial conditions were taken from a 20-years spin-up simulation driven by
ERA5 data, so that the deeper ocean layers could adapt to the present-day climate (S. Grayek,
pers. comm., 2023). Note that for the evaluation of results, we neglected the year 2010 to have
an additional spin-up where NEMO could adapt to the specific transient conditions within each
of the two experiments. For the German Bight, this spin-up of one year is sufficient as the
residence time of water may comprise only up to four months (Becker et al., 1999). In the two
experiments, the daily riverine inflow into the ocean was taken from the uncorrected and bias
corrected discharges of HDW, which were converted to the NEMO grid using a procedure of
Nguyen et al. (2024). For each HD model river mouth box, we associated the nearest coastal
ocean box on the NEMO grid if such a box was found within a search radius of 200 km. Such
a large radius is necessary because the NEMO coastline is very smooth, so many estuaries and
bays in the HD model grid are not resolved by NEMO. If no ocean box was found, the
corresponding HD model box was not linked. Consequently, the simulated discharge data at the
river mouths were placed as freshwater inflow into the corresponding NEMO grid boxes.

## 2.7 Evaluation metrics

The evaluation of the simulated discharge was performed for the grid boxes corresponding to
the discharge station locations within the river network. For the evaluation at these station
locations, we used the mean bias, the Pearson correlation coefficient and the Kling-Gupta
efficiency (KGE; Gupta et al., 2009; Kling et al., 2012). All metrics were calculated with

simulated and observed daily discharge time series for the period considered, using only those days for which observed data are available. The KGE is a quality metric combining bias, correlation and coefficient of variation. If a simulated discharge time series has a KGE > -0.41, then it is a better representation of the observations than the use of the observed long-term mean discharge (Knoben et al., 2019). Note that many ocean model applications still use the latter method.

For the evaluation of simulated salinity in the NEMO experiments, we used daily values and considered

- the mean bias
- the correlation of simulated and observed time series expressed by the Pearson correlation coefficient
- the variability ratio defined by the ratio of the simulated and observed coefficients of variation
- the normalized root-mean-square-error (RSME)
- the centered RSME.

The first four metrics are described, e.g., in Hagemann et al. (2020), while the centered RSME is described, e.g., in Taylor (2001).

## 3 Evaluation of the bias correction

Below, various metrics have been calculated at the station locations and at the river mouths. However, these measures have been assigned to the respective catchment areas for the purpose of graphical presentation.

### 3.1 Evaluation of simulated discharge

The distribution of bias and KGE for HDG and HDW during 1979-2014 (Figure 5) is quite similar to the pattern shown by Hagemann and Stacke (2022) for the ERA5-based discharge. For both simulations, the general discharge behaviour is well captured (KGE > 0.4) for many European rivers, especially in Northern Iberia, Western and Central Europe, and over Northern Russia (Figure 5, lower row). As expected (cf. Hagemann et al., 2020), larger deviations of the simulated from observed discharges occur for rivers that are heavily influenced by human activities such as water abstraction, e.g. for irrigation, and regulation, e.g. by dams. This is the case for many Scandinavian and Turkish rivers as well as the Volga and Don.

In general, the HDW discharges are slightly drier than the HDG discharges, as indicated by larger dry biases in Northern Europe and smaller wet biases in Central Europe. Despite the differences in bias distribution, the KGEs of HDW are similar to or slightly better than those of HDG. Compared to the ERA5-based discharge of Hagemann and Stacke (2022), HDW tends to have smaller discharge biases and better KGEs. This is an expected behaviour caused by the application of a bias correction methodology to the ERA5 data in the generation of the WFDE5 data (cf. Sect. 2.1). An exception to this general improvement occurs over Northern Europe, where the dry bias of HDW tends to be slightly larger and the KGEs lower. Note that Hagemann and Stacke (2022) attributed the dry bias over Northern Europe to an overestimation of the evapotranspiration simulated by HydroPy.

We also note the large-scale patterns of positive and negative discharge biases (Figure 5). Abrupt changes in bias behaviour along the same coastline are rare. Most of the few cases can be attributed to large human water abstractions from the river, i.e. especially for the Ebro River

(see also Section 3.3) and in Turkey, which are not considered by the model. This supports our
assumption about the spatial transferability of the three-quantile bias correction factors. The
bias patterns are related to biases in the atmospheric forcing dataset or biases introduced by the
HydroPy model.

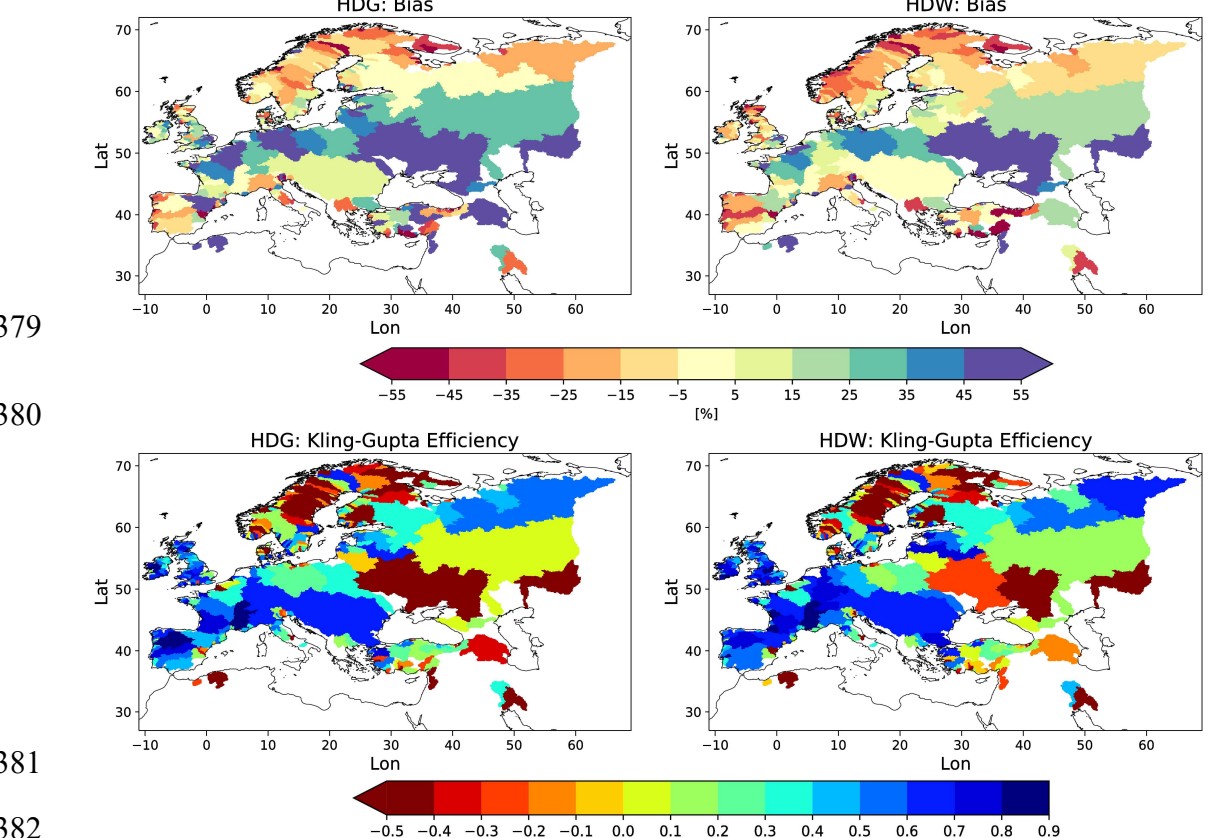

**Figure 5.** Mean discharge bias [%] (upper row) and KGE (lower row) for HDG (left) and
HDW (right) during 1979-2014.

In order to analyse how much the bias correction affects the daily sequence of river runoff
at the station locations, we calculated the correlation between the simulated discharges and the
observations. Supplementary Figure S1 shows that the correlation patterns of HDW and HDW-
BC with observed discharges are quite similar. For rivers where differences can be identified,
the correlation mostly increases for HDW-BC. The correlation between HDW and HDW-BC
is generally higher than 0.95, and only a very few rivers show correlations lower than 0.9. These
rivers are usually rivers that are heavily influenced by human activities, such as the Volga and
the Luleaelven.

## 3.2 Added value of the three-quantile bias correction

In this section, we consider the effect of the bias correction at the station locations and
investigate whether the three-quantile bias correction adds value compared to using only the
mean bias correction. For this purpose, we use HDW and the period 1979-2014.
Both bias correction methods reduce the mean discharge bias to zero or close to zero in the
case of the three-quantile bias correction due to the smoothing around the percentile range
thresholds (see Table 3 for selected rivers). When the mean bias correction is applied, the KGEs
(Figure 6 – left panel) are noticeably improved over Western and Central Europe. However,
with a few exceptions, the KGE pattern over Northern Europe and other areas remains largely
unchanged. This indicates that a correction of the long-term bias in the annual mean discharge
over these areas is not sufficient. Only with the three-quantile bias correction does the KGE
(Figure 6 – right panel, Table 3 for selected rivers) improve considerably over these areas, with
the largest improvements occurring over Scandinavia. The three-quantile bias correction also
leads to some further improvements over Western and Central Europe, where the bias corrected
discharge with the mean bias correction already shows relatively high KGE values, e.g. for the
rivers Elbe, Rhine and Weser.
**Table 3.** Mean bias and KGE of simulated (HDW) and bias corrected discharge during
1979-2014 for selected rivers, where the three-quantile bias correction led to a
noticeable KGE improvement in comparison to the mean bias correction.

| | HDW | | Mean Bias corr. | | 3-quantile Bias corr. | |
|---|---|---|---|---|---|---|
| **River** | **Bias** | **KGE** | **Bias** | **KGE** | **Bias** | **KGE** |
| **Dalaelven** | -32.02 % | -0.32 | 0 % | -0.28 | 0.01 % | 0.48 |
| **Elbe** | 36.44 % | 0.46 | 0 % | 0.60 | -0.06 % | 0.85 |
| **Indalsaelven** | -19.32 % | -0.79 | 0 % | -0.78 | -0.02 % | 0.38 |
| **Odra** | 41.30 % | 0.14 | 0 % | 0.25 | 0.01 % | 0.75 |
| **Rhine** | 14.60 % | 0.74 | 0 % | 0.78 | -0.02 % | 0.85 |
| **Weser** | 33.15 % | 0.55 | 0 % | 0.70 | -0.01 % | 0.90 |



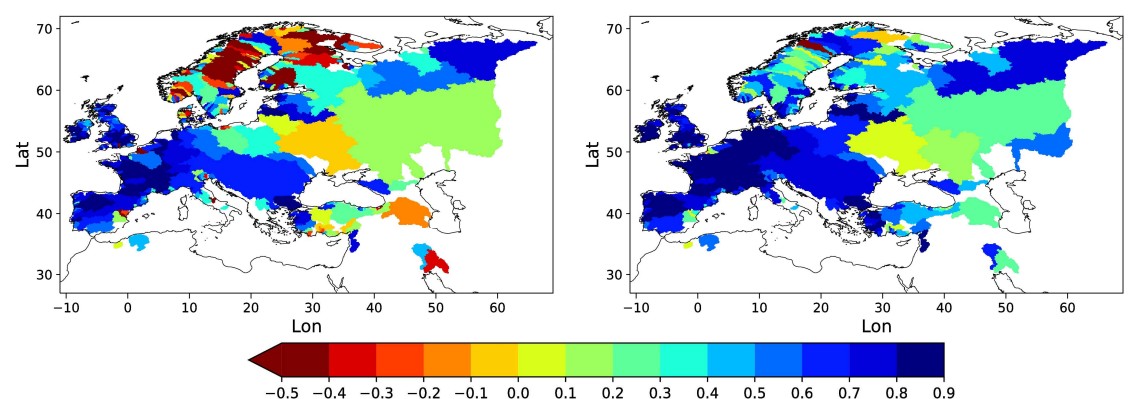

**Figure 6.** KGE for bias corrected HDW discharges using the mean bias correction (left)
and the three-quantile bias correction (right) during 1979-2014.
To visualise the effect of the three-quantile bias correction on the simulated daily discharges,
we consider the corresponding discharge curves for the period 2000-2009 for selected large
rivers. The respective biases and KGE are shown in Table 3 for the period 1979-2014. For the
rivers, Elbe, Weser and Oder, the peak discharges are generally overestimated, while the low
flows are close to the observed values (Figure 7a,c,d). The correction of the high percentiles
leads to a considerable improvement in the representation of the peak discharges, while the
change in the low flows is rather small. The discharge of the Rhine (Figure 7b) is well
represented by HDW. However, the small downward correction of the peak discharges and the
slight increase in the low flows still lead to an improved discharge curve, which is also indicated
by the increased KGE (Table 3).

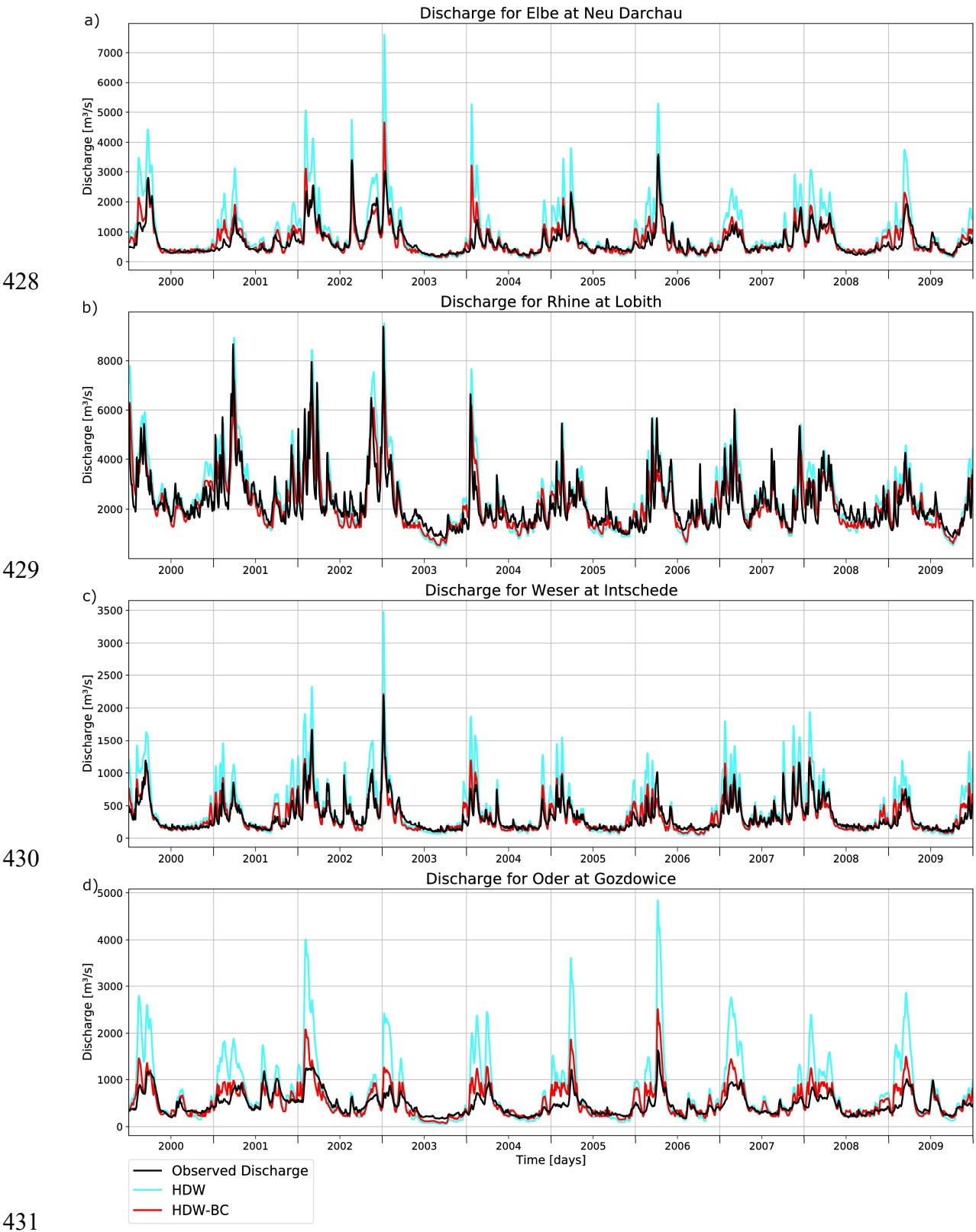





**Figure 7.** Observed and simulated daily discharges for the rivers a) Elbe, b) Rhine,c) Weser and d) Odra during 2000-2009.

As mentioned above, the greatest improvements from the three-quantile bias correction compared to the application of the mean bias correction occur over Scandinavia. Here many

rivers are highly regulated. For this reason, the discharge curves of the Daleaelven and Indalsaelven rivers are examined in more detail in Figure 8. The observed discharges clearly show the effect of the human regulation, where regulation leads to the elimination of peak discharges, while maintaining certain flows during low flow periods. Figure 8 shows that, on the one hand, peak discharges are often suppressed or reduced, especially during the spring, and that, on the other hand, low-flow periods are either almost absent (especially for the Indalsaelven) or show a rather noisy, unnatural daily variability. Here, the bias correction partially mimics these regulation effects by reducing the peak discharges and increasing the low flows.

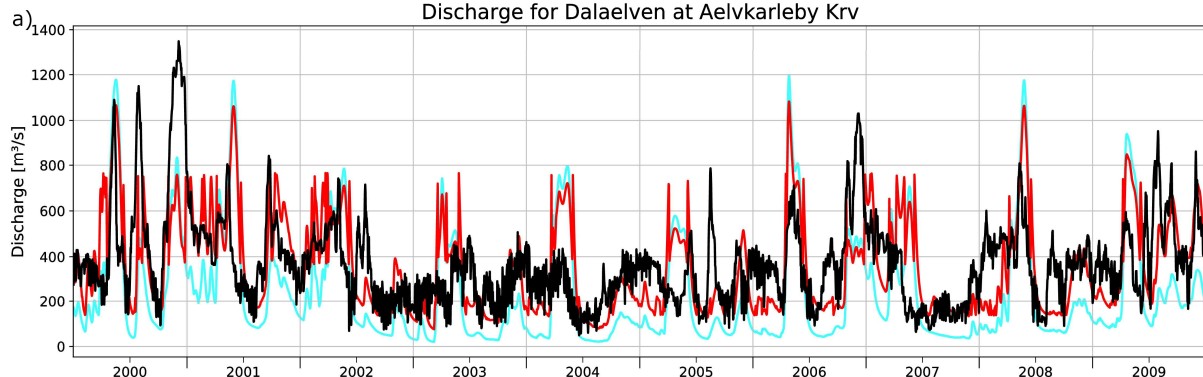

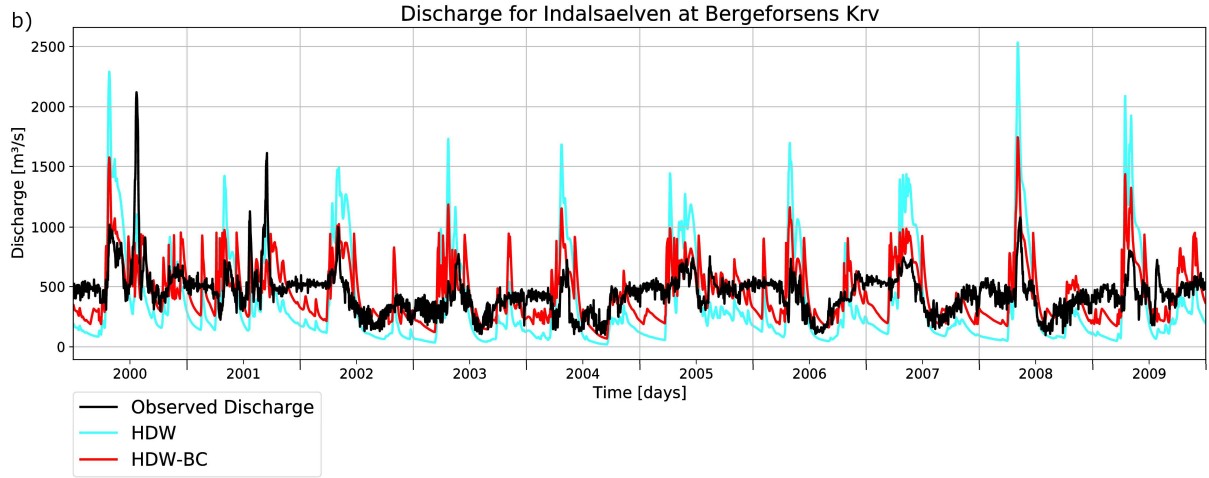

**Figure 8.** Observed and simulated daily discharges for the rivers a) Dalaelven and b) Indalsaelven during 2000-2009.

## 3.3 Application of the bias correction for a different time period

To consider the effect of the bias correction for the applications over different time periods, we derived bias correction factors for HDG during 1979-2014 and applied the factors for the period 1950-1978.

For HDG, the distributions of bias and KGE are quite similar between the two periods 1950-1978 (Figure 9 – left column) and 1979-2014 (Figure 5 – left column). Consequently, the bias correction leads to similar improvements in the KGE (Figure 9) as for the most recent period (not shown). The bias also becomes small for most of the rivers. Noticeable exceptions are the Dnjepr, Volga and some rivers in Southern Europe. This may be related to differences in the

anthropogenic influence on the discharge between the two periods, as is the case for the river
Ebro. Here, the large wet bias (51.65 %) in the more recent period is contrasted with a small
wet bias (12.05%) in the earlier period (Figure 10). Since large anthropogenic water
abstractions occur in the Ebro River (Merchán et al., 2013), this seems to be related to the
different irrigation activities in the two periods, which are much more pronounced in the more
recent years. The latter can be seen by looking at the observed daily discharges between 1960-
1969 and 2000-2009 (Figure 10). In the earlier period, the Ebro discharge still shows some
variations according to the sequence of weather events in the dry season. However, in the later
period, the observed discharge includes only very small variations during the dry season,
indicating more intense human water abstraction than in the earlier period. Consequently, the
bias correction based on the recent wet bias leads to a dry bias (-25.78 %) in the corrected Ebro
discharge in the earlier period. However, the KGE decreases only slightly from 0.68 to 0.63, so
that the deterioration of the mean bias seems to be largely compensated by the correction of the
different percentile ranges.

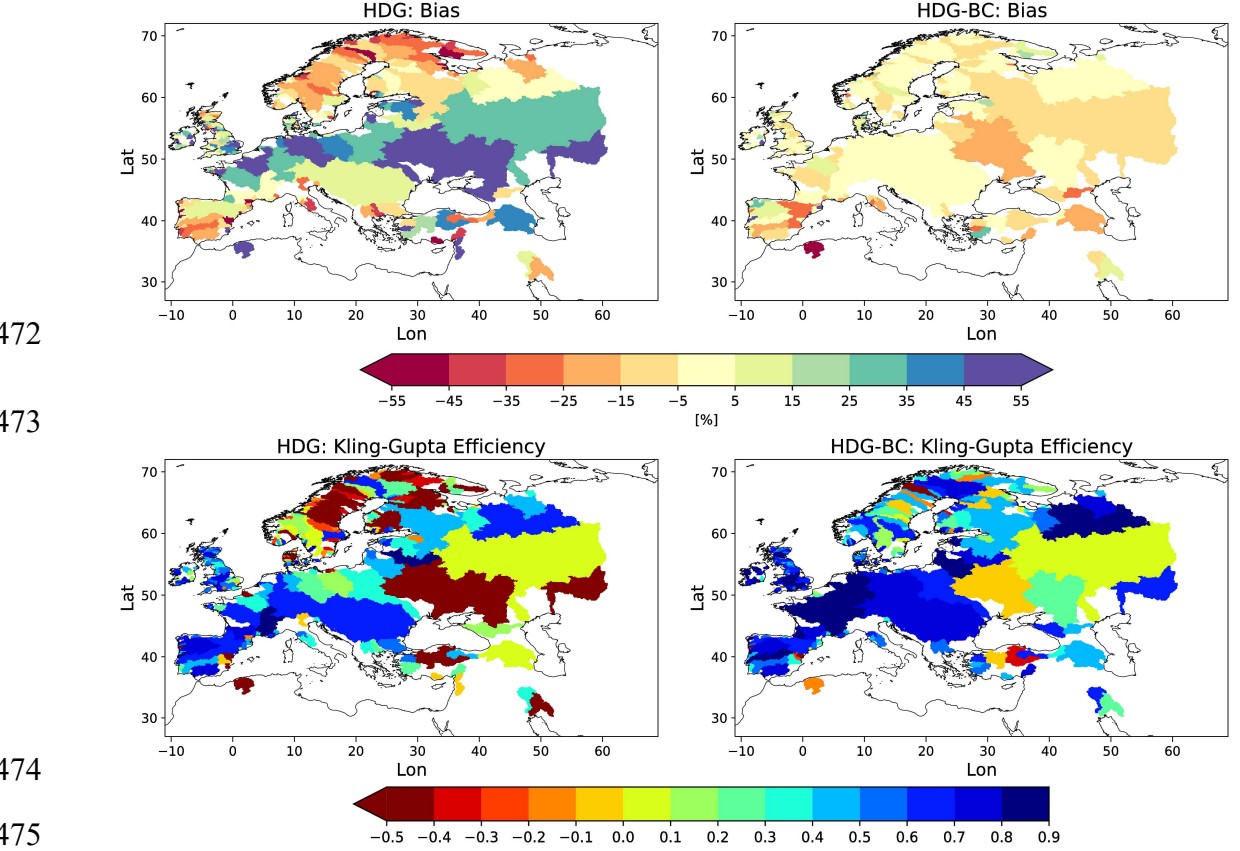

**Figure 9.**  Mean discharge bias [%] (upper row) and KGE (lower row) for HDG (left) and
HDG-BC data (right) during 1950-1978.

## 3.4 Effect of the bias correction on contemporary trends

As mentioned in Sect. 2.4, our three-quantile bias correction is similar to a very coarse quantile
mapping, and quantile mapping has been flagged as potentially not suitable for climate
simulations as it has been shown to modify trends (e.g. references in Cannon et al., 2015).
However, Maraun et al. (2017) pointed out that a debate has arisen about whether trend
modification by variance-adjusting bias correction methods actually improves or degrades the
raw climate change signal. They further argued that purely statistical arguments cannot resolve

this issue, which requires process understanding. With respect to runoff, the latter needs to take into account spatial and temporal characteristics of rivers and events, which is beyond the scope of the present large-scale study.

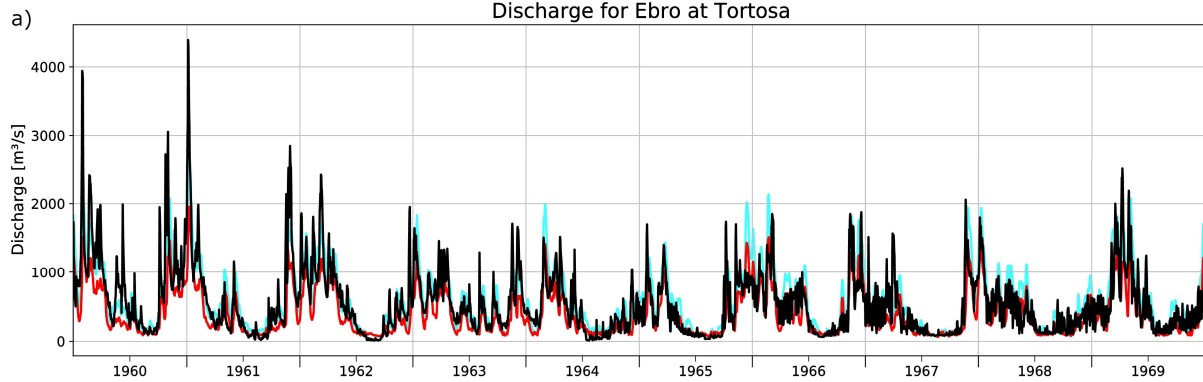

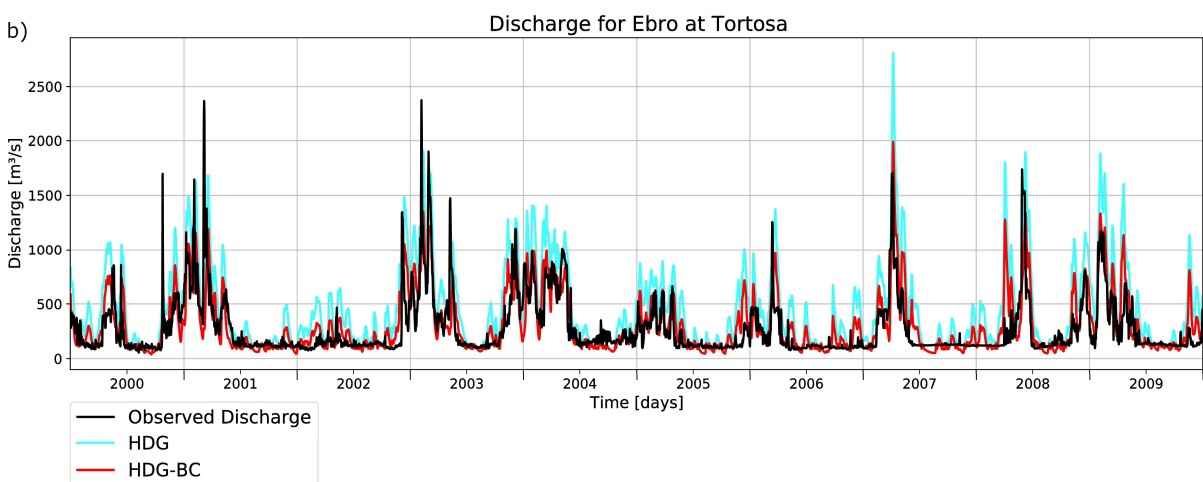

**Figure 10.** Observed and simulated daily discharge based on HDG for the Ebro river during a) 1960-1969 and b) 2000-2009.

To investigate the effect of the bias correction on contemporary trends, we calculated trends in the annual maximum, mean and minimum discharge for the period 1979-2014 and compared the results for HDW and HDW-BC (Figure 11). The trend patterns are generally within the range spanned by the two datasets considered in Hagemann and Stacke (2022). For the annual maximum and mean discharge, the trend patterns are only slightly changed by the bias correction. For the annual minimum discharge, the trend pattern is quite similar in HDW and HDW-BC. However, there are a few more rivers where the magnitude of the trend is affected by the bias correction. This is particularly the case over Scandinavia where many rivers are regulated, so that that the correction of the low percentile range is often strong to account for the effect of regulation on low flows (cf. Sect. 3.2).

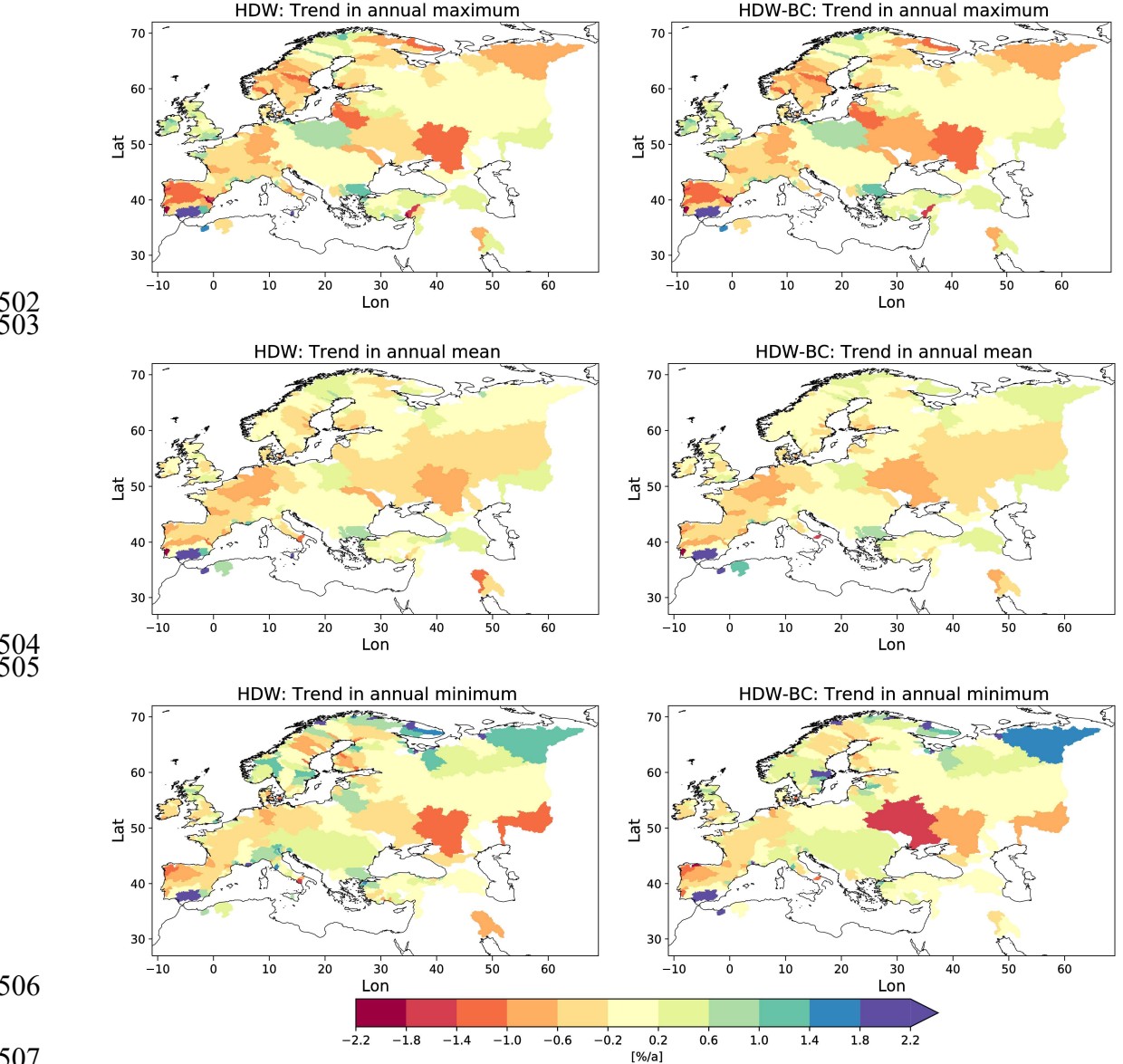

**Figure 11.** Trends in annual maximum (1st row), mean (2nd row) and minimum (3rd row) discharge [%/a] for HDW (left column) and HDW-BC (right column) from 1979-2014.

## 4 Evaluation of the inflow into sea basins

To evaluate the simulated and bias corrected discharges at the river mouths, we considered the integrated inflow into different sea basins. First, we evaluated the discharges into the Baltic Sea using HELCOM and IGC-EMO data in Section 4.1. We then compared the discharges to the North Sea and the Northeast Atlantic with OSPAR and up-scaled (see Section 2.5) IGC-EMO data in Section 4.2.

### 4.1 Baltic Sea

In order to achieve a maximum overlap of the simulated discharge time series data with the HELCOM data (cf. Section 2.5), we considered 1995-2019 as the evaluation period for the Baltic Sea and its seven sub-basins (Figure 4 – upper panel). For the whole Baltic Sea and most of its sub-basins, the bias correction improves the basin inflows if compared to the HELCOM estimates (Table 4, Figure 12). Only for the Gulf of Finland and the Gulf of Riga, the bias

correction leads to a slightly larger bias while the biases of HDW in these basins are relatively
small. When the simulated inflows are compared with the IGC-EMO estimates, similar results
are obtained, except for the Gulf of Riga. Here, the IGC-EMO estimates are about 32% larger
than the HELCOM estimates, indicating a larger uncertainty in at least one of these two
estimates. For the Gulf of Riga basin, the major part of the inflow is contributed by the Daugava
river. In IGC-EMO, the discharge from the Daugava is based on observed time series from
1970-1990. These data were extended to earlier and later periods, e.g. by climatological values
and trend preservation (Van Leeuwen and Hagemann, 2023). For 1970-1990, the mean IGC-
EMO discharge comprises 623 m³/s at the Daugava mouth, while this has increased by ca. 45%
in 1995-2019 (902 m³/s). However, this strong increase cannot be seen in the observed
discharge time series at the station Daugavpils that covers about three quarter of the Daugava
catchment. Here, the discharge increases only slightly from 1970-1999 (439 m³/s; 95%
temporal data coverage) to 1995-2019 (452 m³/s; 83% temporal data coverage). This indicates
a large overestimation of the IGC-EMO Daugava discharge during 1995-2019 and, hence, also
of the respective Gulf of Riga inflow.

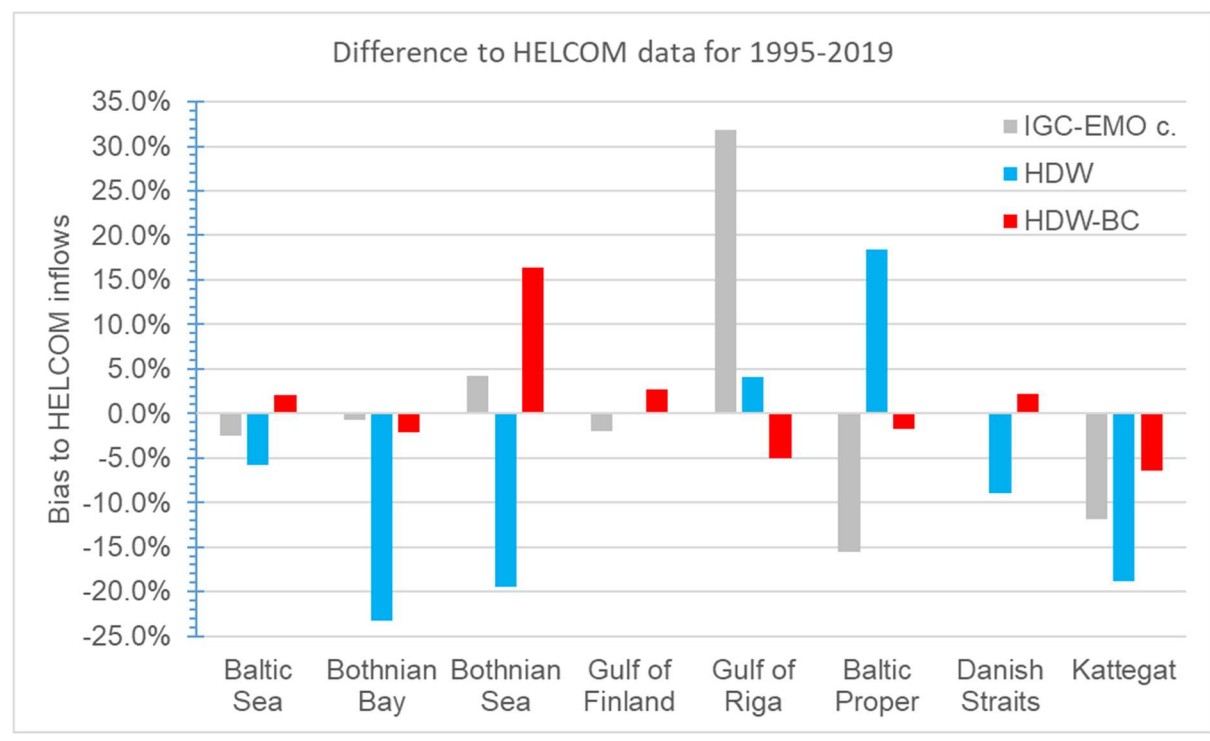


**Figure 12.** Relative difference in basin inflows compared to HELCOM data for 1995-
2019. Note that no IGC-EMO estimate is provided for the Danish Straits as the
respective river catchment coverage in IGC-EMO is too small.

**Table 4.** Estimated and simulated inflows (unit: m³/s) into the Baltic Sea and its major
sub-basins during 1995-2019. Note that for the Danish Straits no IGC-EMO estimate is
provided as the respective catchment area coverage of the rivers in IGC-EMO is too low.

| Sea basin | HELCOM | IGC-EMO c. | HDW | HDW-BC |
|---|---|---|---|---|
| Baltic Sea | 15676 | 15286 | 14764 | 15995 |
| Bothnian Bay | 3444 | 3420 | 2642 | 3369 |
| Bothnian Sea | 2913 | 3038 | 2347 | 3391 |
| Gulf of Finland | 3519 | 3448 | 3520 | 3612 |

| | | | | |
|---|---|---|---|---|
| Gulf of Riga | 1071 | 1411 | 1114 | 1017 |
| Baltic Proper | 3436 | 2901 | 4070 | 3377 |
| Danish Straits | 217 | 0 | 198 | 222 |
| Kattegat | 1077 | 949 | 873 | 1008 |

## 4.2 North Sea and Northeast Atlantic

Due to the different treatment of unmonitored regions by the OSPAR countries (cf. Section 2.5), and thus of the respective sea basin areas, we have not corrected the OSPAR inflows. Instead, we have also considered up-scaled IGC-EMO data as alternative estimates of basin inflow (as in Section 4.1). Table 5 shows simulated and estimated basin inflows for the considered OSPAR regions (cf. Figure 4 – lower panel). Note that IGC-EMO data for the Norwegian shares of the Barents Sea, Norwegian Sea and North Sea, and the North Spanish Atlantic are not included in the following comparisons due to their limited area coverage. When comparing the simulated sea basin inflows with the OSPAR and IGC-EMO data, we found that the bias correction improves the simulated inflows for most of the OSPAR regions (Figure 13). Exceptions are the values for the Celtic Sea and the Irish Sea. For the Celtic Sea, the bias corrected inflows are very close to the uncorrected inflows and the difference to the OSPAR data is rather small. For the Irish Sea, the bias corrected inflows are somewhat larger than the uncorrected ones, with both showing large differences (52.5% and 47.5%) to the OSPAR data. Here both inflows are closer to the IGC-EMO estimate, which exceeds the OSPAR estimate by about 40%.

**Table 5.** Estimated and simulated inflows (unit: m³/s) into major sub-basins of the North Sea and the Northwest Atlantic during 1961-1990. Note that the North Sea does not comprise Skagerrak and the English Channel. Up-scaled IGC-EMO basin estimates for which the fractional catchment coverage (see Table 1) of IGC-EMO rivers is less than 75% are considered as highly uncertain and are therefore only given in brackets (cf. Sect. 2.5). The same applies to the OSPAR inflow into the Northern Spanish Atlantic.

| Sea basin | OSPAR | IGC-EMO c. | HD | HD-BC |
|---|---|---|---|---|
| North Sea | 9190 | 6600 | 9798 | 9164 |
| Norwegian North Sea | 3534 | (1499) | 2038 | 2856 |
| Norwegian Barents Sea | 2294 | - | 1106 | 1723 |
| Norwegian Sea | 3663 | - | 2242 | 2922 |
| Skagerrak | 2544 | 2113 | 1956 | 2292 |
| German Bight | 1344 | 1505 | 2025 | 1419 |
| English Channel | 1250 | 1011 | 1498 | 1222 |
| Celtic Sea | 976 | 839 | 1016 | 1016 |
| Irish Sea | 672 | 939 | 992 | 1025 |
| French Atlantic | 2862 | 2391 | 3147 | 2684 |
| Northern Spanish Atlantic | (359) | (1655) | 1104 | 1550 |

While the OSPAR values from Ireland include estimates for unmonitored areas, this is not the case for the United Kingdom (Table 2). Farkas and Skarbøvik (2021) list the rivers

contributing to the OSPAR value (560 m³/s) from the United Kingdom part of the Irish Sea
catchment (35000 km²). Adding up the catchment areas of each river, obtained from various
online resources, gives a coverage of about 70%. In order to account for this under-
representation of the catchment area, an up-scaling can be performed, similar to the treatment
of the IGC-EMO data. This would give an estimate of about 803 m³/s for the Irish Sea inflow
from the United Kingdom and thus 915 m³/s for the whole Irish Sea. The respective IGC-EMO
inflow is close to this value (+2.6%) and the overestimation of inflows is less pronounced for
HD and bias corrected discharges with +8.4% and +12% respectively.

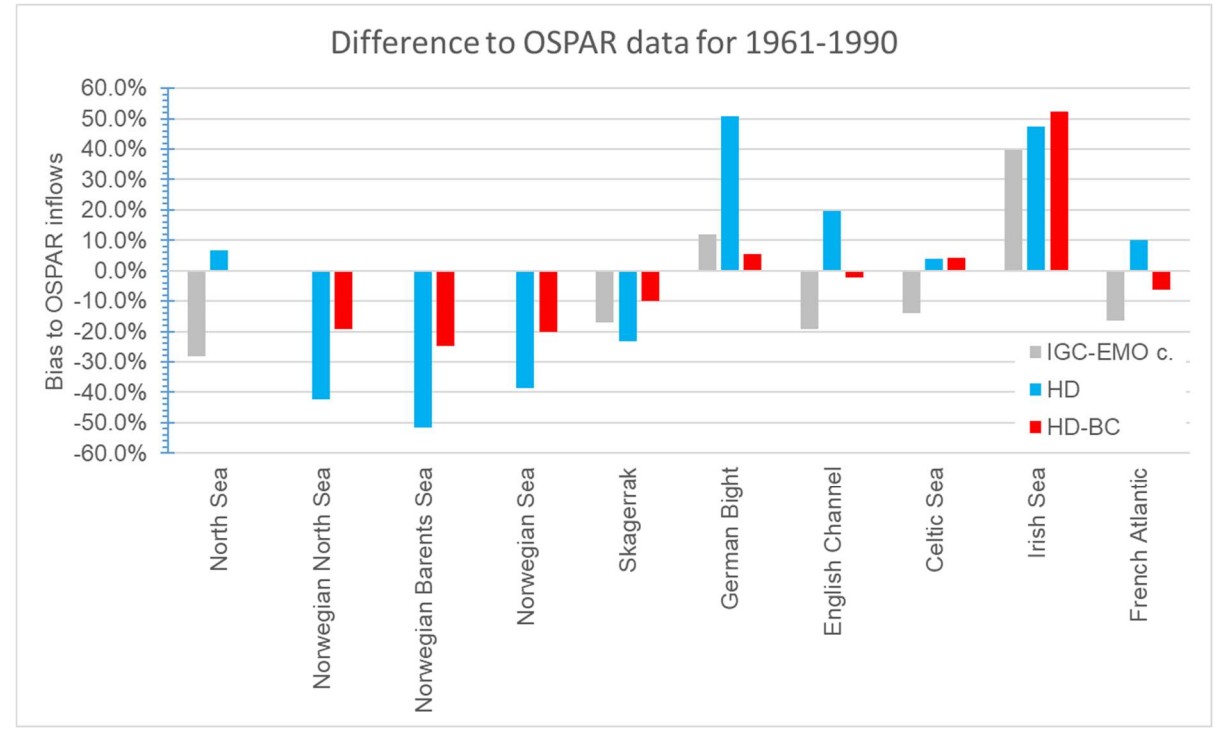


**Figure 13.** Relative difference in basin inflows compared to OSPAR data for 1961-1990.
IGC-EMO basin estimates for which the fractional catchment coverage (see Table 1)
is less than 75% are not shown.
## 4.3 Simulated salinity in the German Bight
Using the two experiments described in Sect. 2.6, we evaluated the simulated sea surface
salinity (SSS) with satellite-based analyses and in-situ observations for the period 2010 to 2018.
The SSS analyses were derived using a multivariate optimal interpolation algorithm that
combines sea surface salinity images from several satellite sources, such as the National
Aeronautics and Space Administration  Soil Moisture Active Passive satellite and the European
Space Agency  Soil Moisture Ocean Salinity satellite, with in-situ salinity measurements
(Droghei et al., 2018). These SSS data are available with a spatial resolution of 0.125°.
Figure 14a shows the mean analysed SSS in the German Bight for the period 2010-2018,
with lower salinities near the west coast of Germany and higher salinities towards the west. The
NEMO simulation using the uncorrected discharges of HDW (Figure 14c) has too low SSS in
coastal areas, especially near the estuaries. This low bias is reduced using the bias corrected
discharges (Figure 14d), as the general effect of the bias correction in the German Bight leads
to reduced riverine inflows (cf. Figure 13) and hence increased SSS in coastal areas (Figure
14b). Similar improvements can also be seen in June 2013 when the Elbe flood is strongly
influences the SSS of the German Bight (Figure S2). Here, the increase in salinity due to the
bias corrected runoff (Figure S2b) is more pronounced than in the long-term mean (Figure 14b).
In addition, we found that use of the bias corrected river runoff also improves the SSS variability
expressed by its coefficient of variation, shown in Figure 15.

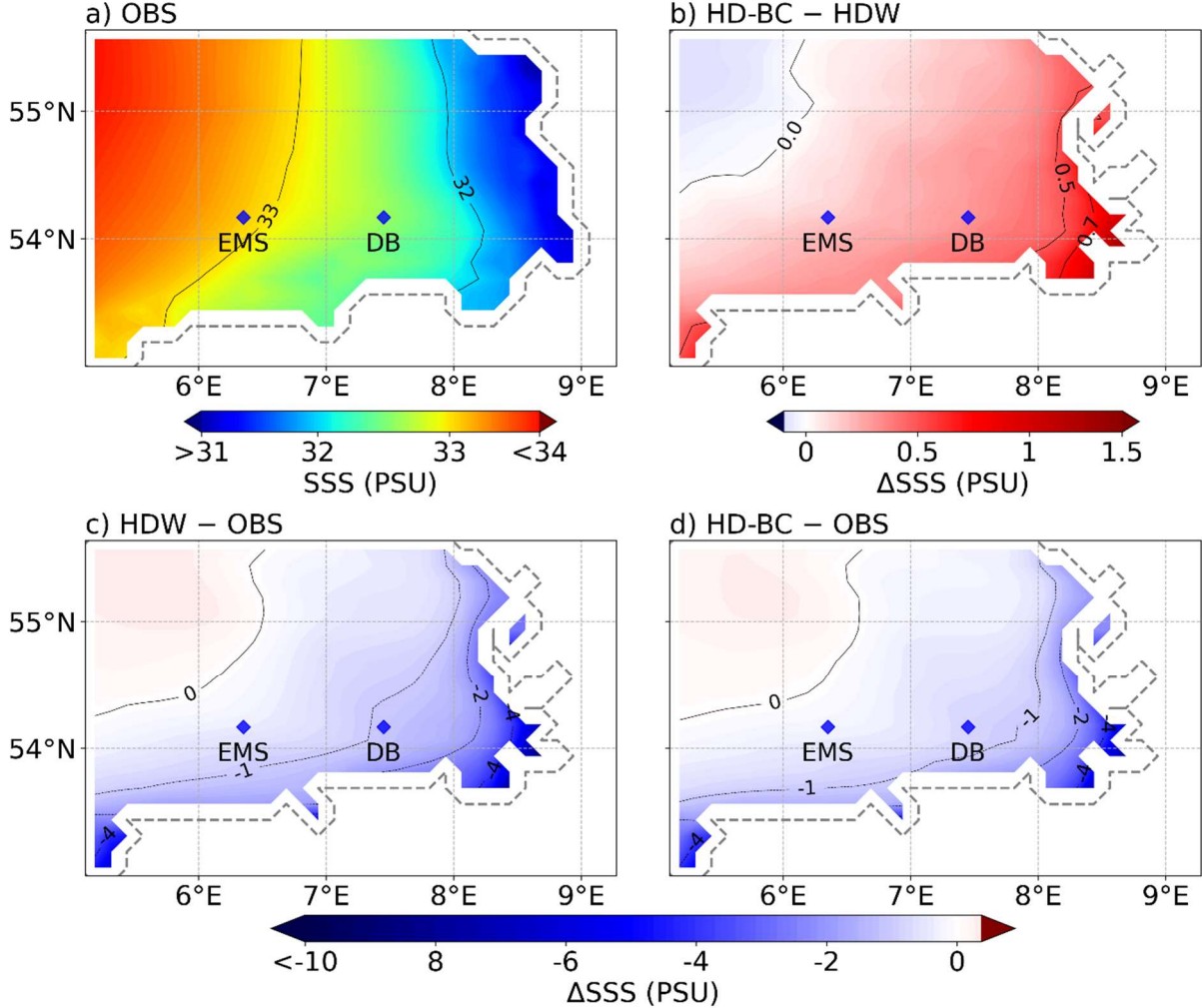


**Figure 14.** Mean analyzed SSS: a) Droghei et al. (2018) data (OBS) and various SSS
604          differences of the NEMO experiments in the German Bight for the period from 2010
605          to 2018. The SSS differences comprise b) HD-BC minus HDW, c) HDW minus OBS,
606          and d) HD-BC minus OBS.


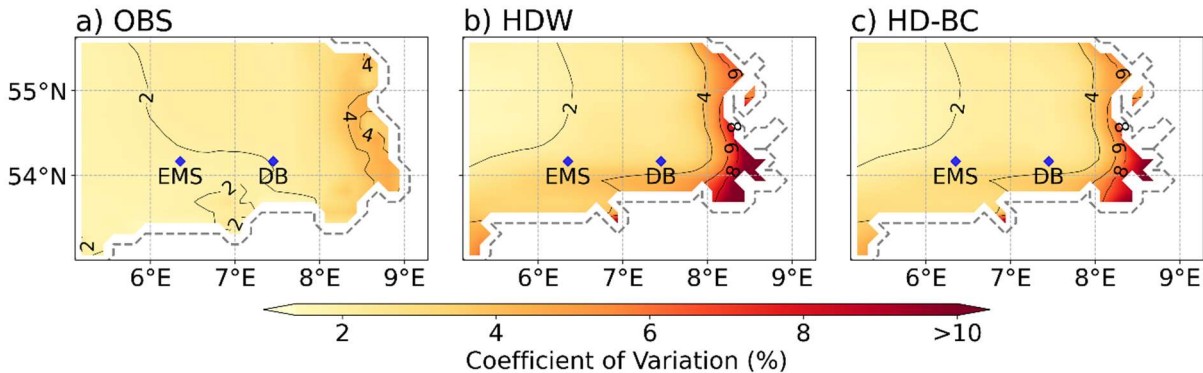

**Figure 15.** Coefficients of variation of SSS in the German Bight for the period from 2011-2018: a) OBS, b) HDW and c) HD-BC

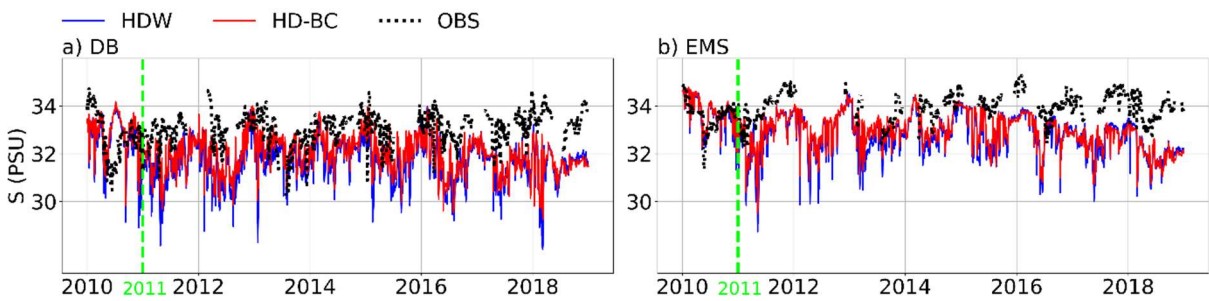

**Figure 16.** Observed (OBS) and simulated daily time series of salinity in 6 m depth for the stations a) *Deutsche Bucht* (DB) and b) *EMS*. Unit: PSU. The blue and red solid lines correspond to the HDW and HD-BC experiments, respectively. The green line separates the spin-up period in 2010 from the evaluation period 2011-2018.

In addition, we had access to salinity measurements at two stations in the German Bight operated by the German Federal Maritime and Hydrographic Agency as part of the Marine Environmental Monitoring Network in the North and Baltic Seas. These two stations are *Deutsche Bucht* (DB; located at 54.17ºN, 7.45ºE) and EMS (54.17ºN, 6.35ºE) and their locations are shown in Figure 14. In general, the bias corrected discharges lead to improved characteristics of the daily salinity at 6 m depth at the *Deutsche Bucht* and *EMS* stations (Figure 16, Table 6). Here the bias, normalized and centred RSME are decreased, and the coefficient of variation is closer to the salinity observations for HDW-BC. This means that the bias correction improves the mean and the variability of the simulated salinity at these stations. However, the correlation with the observed salinity measurements is somewhat reduced. Note that temporal SSS variations are strongly influenced by local currents, vertical mixing and wind-wave-surface interactions. Therefore, signals from an improved river runoff can easily be obscured by the noise from these processes, which can also differ at the point scale of the station measurements and at the grid scale of the respective NEMO grid box. This is reflected in the relatively low correlation values. Furthermore, this can be seen when the gridded SSS data of Droghei et al. (2018) are used as a reference for the metrics at the station locations (Supplementary Table S1). Here, all metrics improve with HDW-BC, even the correlation. However, the correlation is lower than with the station observations, which is also the case for the correlation of the gridded SSS data itself with the station observations (*Deutsche Bucht*: 0.15; *EMS*: 0.18). Considering only the year 2013, when the influence of the Elbe flood on the salinity at the *Deutsche Bucht* station is more pronounced (Nguyen et al., 2024), the correlation also improves when using

HDW-BC for both references (Table 6 and S1). It seems that in NEMO the positive effect of
using bias corrected discharges is limited to near-surface salinities, as there is no noticeable
effect at 30 m depth (not shown). This is consistent with the fact that the *Deutsche Bucht* and
*EMS* stations are located in an area where the salinity is temporarily stratified, depending on
the meteorological conditions and the intensity of river runoff (Klein and Frohse, 2008).
In summary, the results of the NEMO experiments indicate the beneficial effect of using bias
corrected discharges on the simulated SSS in coastal areas. However, although the low SSS
biases are reduced by using the bias corrected discharges, the simulated SSS is still
underestimated in coastal areas, especially close to the estuaries of large rivers (Figure 14d).
This may be attributed to the rather smooth coastline of the NEMO ocean grid. Here, most parts
of the large estuaries of the rivers Elbe, Ems and Weser are not included. In reality, a major part
of the mixing of the riverine freshwater inflow and the saline North Sea happens within these
estuaries. In the NEMO model setup, the freshwater inflow is introduced at the respective river
mouth points of the smooth NEMO coastline where it starts to mix with the saline North Sea
water. Consequently, the simulated water at and near those points is much fresher than in reality,
which leads to the low SSS bias. Note that on the one hand such a smooth coastline is necessary
in NEMO to avoid numerical instabilities. On the other hand, the spatial resolution of the
NEMO grid is not high enough to adequately resolve parts of the longer estuaries.
**Table 6.**    Various metrics (see Sect. 2.7) of the simulated salinity time series in 6 m
656       depth compared with the observations at the stations *Deutsche Bucht* and *EMS* for
657       2011-2018 and at *Deutsche Bucht* for 2013.

| | 2011-2018 | | | | 2013 | |
| | *Deutsche Bucht* | | *EMS* | | *Deutsche Bucht* | |
| Metric | HDW | HDW-BC | HDW | HDW-BC | HDW | HDW-BC |
|---|---|---|---|---|---|---|
| Bias [%] | -4.5 | -3.7 | -4 | -3.6 | -3.2 | -1.8 |
| Variability ratio [%] | 142.7 | 125 | 151.5 | 136.1 | 82.9 | 74.2 |
| Normalized RMSE [%] | 40.1 | 34.3 | 51.3 | 47.6 | 36.2 | 27.1 |
| Centered RMSE | 0.94 | 0.89 | 0.73 | 0.72 | 0.97 | 0.89 |
| Correlation | 0.24 | 0.21 | 0.48 | 0.39 | 0.20 | 0.28 |


## 5   Summary and Conclusions

In the present study, we have introduced a methodology for the bias correction of European
river runoff to provide corrected riverine inflows as forcing for ocean models in offline and
coupled system model simulations. The central part of this methodology is a three-quantile bias
correction, which can correct different biases for low, medium and high discharges. The bias
correction parameters are derived in two steps. First, different correction factors for low,
medium and high flows are derived for each river considered (cf. Sect. 2.5) at the location of
the most downstream station for which daily discharge measurements were available. These
factors were then transferred to the respective river mouth on the HD model grid and to adjacent
coastal inflow points in its vicinity.
The evaluation of the bias corrected discharge at the station location showed that the bias
correction greatly improved the simulated discharges. For the evaluation of the bias corrected
discharge at the downstream station locations, we considered the mean bias and the KGE, which
is a quality metric combining bias, correlation and coefficient of variation. Considering the
same period as used to derive the bias correction factors, the mean bias is trivially close to zero.
However, the bias is also substantially reduced for most rivers if a different period is considered.
Irrespective of the period, the KGE pattern generally improves for the bias corrected discharges
and shows high values for many rivers. Exceptions are those rivers with a very strong
anthropogenic distortion of the natural flow, e.g. by many dams or large water withdrawals.
Here, despite of some improvements, the KGE values are still rather low, such as for the rivers
Dnjepr, Volga, Luleälven and a few Turkish rivers flowing into the Black Sea. The KGE also
shows the beneficial effect of the three-quantile bias correction, as correcting only the long-
term mean annual discharge bias is not sufficient in many areas, especially in northern Europe.
We found that the three-quantile bias correction often improves the KGE in regulated rivers, so
that it appears to mimic the effect of regulation, where regulation leads to the elimination of
peak flows while maintaining certain flow levels during low flow periods.
The evaluation of riverine inflows to the sea at river mouths with observed daily discharge
is rarely possible as there are usually no river gauges available. Even if there is a gauge at the
mouth of a river, the measurements are often affected by tidal influences from the coast, so that
the measured amounts may not represent the actual river discharge. For obvious reasons, it is
also difficult to compare simulated inflows with observed discharges for unmonitored rivers.
Therefore, we compared the simulated and bias corrected discharges with long-term mean
inflow estimates into different sea basins from HELCOM, OSPAR and IGC-EMO. For most of
the basins considered, the bias correction improves the simulated inflows. This indicates a
reasonable performance of the approach to transfer the bias correction factors obtained at the
downstream stations to the respective river mouths and adjacent coastal areas. The improved
inflows to the sea basins, together with the fact that the discharge bias behaviour tends not to
vary abruptly along the same coastline, underpin the validity of our transferability approach.
Exceptions are the Gulf of Finland, the Gulf of Riga, the Celtic Sea and the Irish Sea. For the
Gulf of Finland and the Celtic Sea, the deviations of the uncorrected and bias corrected inflows
from the inflow estimates are rather small. For the Gulf of Riga, the deviations of the
uncorrected and bias corrected inflows from the HELCOM estimates are also small, but they
significantly underestimate the IGC-EMO estimates. However, this could be due to a large
overestimation of the Daugava discharge during the period 1995-2019 in the IGC-EMO data
and thus also of the corresponding Gulf of Riga inflow. For the Irish Sea, IGC-EMO seems to
be closer to reality as the OSPAR inflow does not cover the unmonitored rivers in the British
part of the catchment.
A caveat applies for rivers where the human influence on river flow has changed
significantly over time. Applying bias correction factors derived for 1979-2014 to earlier
periods may lead to errors for regulated rivers in years before these regulatory measures were
implemented. This is the case for the Ebro, where irrigation activities have largely intensified
during the period 1979-2014 compared to earlier periods (see Sect. 3.3). A detailed analysis of
the rivers and periods concerned is beyond the scope of this study. However, at least for the
period 1950-1978, the KGE distribution does not seem to be significantly affected, as there is
no noticeable deterioration.
We have shown that our bias correction method works well for Europe at the station
locations as well as for the riverine inflow into northern and western European sea basins. Using
two NEMO simulations in the German Bight, we have also shown that the use of the bias
corrected discharges as forcing leads to an improved simulation of sea surface salinity in coastal
areas especially regarding the mean salinity and its variability. However, for the potential
transfer of the bias correction methodology to other regions, it has to be pointed out that the

application of the three-quantile bias correction over a region only makes sense if a large part of the catchment area is covered by available daily discharge measurements. As the three-quantile bias correction is based on biases in three percentile ranges of daily flows, it is also suitable for the use in climate change applications. Here the bias correction factors can be derived from a historical discharge simulation and then applied to future projections or past reconstructions. In addition, the bias correction can also be applied in regional coupled system model simulations, where the bias correction factors can be derived from an initial simulation and then applied during the run-time of the actual coupled simulation. This capability has been implemented in the HD model v5.2.2 (Hagemann et al., 2023) and is currently being applied in the coupled system model GCOAST-AHOI (Ho-Hagemann et al., 2020). Finally, we note that the bias corrected discharges are available from the World Data Centre for Climate and are already used within the CoastalFutures project (https://www.coastalfutures.de).

## Data Availability Statement

Many of the observed daily discharge data used can be obtained from the Global Runoff Data Centre (https://grdc.bafg.de/GRDC/EN/02_srvcs/21_tmsrs/riverdischarge_node.html). Other data have been retrieved from public websites associated with the sources referred to in Sect. 2.5. GSWP3 data were retrieved from the ISIMIP data portal (https://data.isimip.org ) and WFDE5 data were retrieved from the Copernicus Climate Data Store (https://cds.climate.copernicus.eu). OSPAR data were taken from an OSPAR report (Farkas and Skarbøvik, 2021) or its associated data available on the OSPAR webpage (https://odims.ospar.org/en/search/?dataset=rid-data-reports). This study has been conducted using E.U. Copernicus Marine Service Information data on SSS (https://doi.org/10.48670/moi-00051) and some French discharge measurements. The daily data of surface runoff and subsurface runoff as well as the simulated and bias corrected discharge data (Hagemann and Stacke, 2023) can be accessed via the World Data Centre for Climate at the German Climate Computing Center.

## Acknowledgments

This study was conducted within the CoastalFutures project that was funded by the German Federal Ministry of Education and Research under grant number 03F0911E. TN was supported by the subproject 'A6 - The earth system variability and predictability in changing climate' of Germany's Excellence Strategy EXC 2037 'CLICCS - Climate, Climatic Change, and Society' with project no. 390683824, funded by the Deutsche Forschungsgemeinschaft (German Research Foundation). We thank the German Climate Computing Center for providing the computing resources to perform the HD simulations. We acknowledge the Copernicus Climate Data Store and the ISIMIP project for making WFDE5 and GSWP3 datasets available. We are deeply indebted to all data providers. We are also grateful to Sonja van Leuwen (Royal Netherlands Institute for Sea Research) for providing us with the latest version of the IGC-EMO data. We are thankful to Sebastian Grayek (Helmholtz-Zentrum Hereon) for the discussion on his NEMO results using an initial version of the bias corrected discharges. Finally, we thank Tobias Stacke (Max Planck Institute for Meteorology) for conducting the HydroPy simulations published in Hagemann and Stacke (2023).

## Author Contributions

SH developed and applied the three-quantile bias correction, conducted the discharge simulations and analysis of results, and wrote the manuscript. HH conducted the NEMO

simulations, helped with the analysis of results and revised the manuscript. TN evaluated the
SSS data of the NEMO simulations, helped with the analysis of results and revised the
manuscript.
**Conflict of Interest Statement**
The authors declare that the research was conducted in the absence of any commercial or
financial relationships that could be construed as a potential conflict of interest.

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
