# Peer review of "A three-quantile bias correction with spatial transfer for the correction of simulated European river runoff to force ocean models"

_EGUsphere, 2024_

## Author Comment (AC1)

**Reply to the comments of reviewer 1**

We thank the reviewer for the thorough reading of the manuscript and the valuable remarks that helped us to improve the manuscript. In the following, the original reviewer comments are given in italic and all line numbers and figure numbers refer to the original submitted version that was reviewed if not mentioned otherwise.

**Reply to review of reviewer 1**

**General comments**

*The article is logically structured, writing quality is clear (although a bit heavy on acronyms), and figures appropriate.*

*The article is submitted to Ocean Science, but the focus is on a hydrology model bias correction method. Freshwater runoff is an important forcing of coastal ocean simulations, and the article includes ocean simulations as a diagnostic. The journal-fit is not bad, but the readers of Ocean Science may be somewhat unfamiliar with hydrology models and water routing. More attention on summarizing the pertinent information for a oceanographer audience would be worthwhile, particularly sections 2.2 and 2.3.*

We added some text in response to your specific comments on line 107-119, 115-117 and 120-143 given below. Please note that we originally planned to add more details in the sections 2.2 and 2.3. However, the editor correctly pointed out that these details comprised a lot of doubling of text with the information provided in Hagemann et al. (2020) and Hagemann and Stacke (2022). Consequently, we removed these details and referred to the respective publications. In addition, we reduced the number of acronyms.

*The primary deficiency in this article is the absence of comparisons with other bias correction methods. Including such comparisons would put the proposed method into perspective in a landscape of methods. This would enable an exploration into it's relative strengths/weaknesses, and reveal scenarios where it should and should not be applied. E.g. does it generally outperform others, or overcome a common deficiency in others, or have better transferability, or work in scenarios where others do not, etc.*

Please see our response to your specific comments to line 53-66, 71-73, 147-163, 174-180 and 593-603 given below.

For a specific river, our new method does not outperform other bias correction methods. However, the advantage of our method is its large-scale applicability for current and future climate conditions, and the transfer of bias-correction factors from station locations to river mouths and neighbouring catchments. To make this clearer, please see our response to 53-66, 174-180.

*A lesser shortcoming is the limited use of the ocean simulations and observational datasets as a check on the bias correction method through nearshore ocean salinity. Improved agreement on an 8-year mean sea surface salinity comparison does not showcase the strengths of this 3-band bias correction method – simple bias correction, or the climatology runoff datasets, should be able to achieve that result. While a reduced RMSE is shown with respect to salinity*

*station data, it is unclear if that is RMSE or CRMSE; the former may be dominated by the bias improvement, while the latter would indicate improved salinity variability. The salinity observations and ocean model data should be examined more closely to look for improvements in the salinity at shorter time scales.*

Please see our response to your specific comments to the line 495-496 and 507-508 where we added analyses and figures.

*Summarizing, the article has potential but needs to showcase the 3-band method in the context of established bias correcting methods. The ocean simulations and salinity observations are underutilized for the purpose of showing improved nearshore salinity following the bias correction; a focus on improved salinity variability would put them to better use and strengthen the findings.*

**Specific comments**

*Lines 53-66: the authors note that quantile mapping is an established bias correction method, providing a couple references where it was used successfully and a couple where it did not perform well. (a) is quantile mapping expected to succeed or fail for the present case of European runoff? Meanwhile, there is no mention of other bias correction methods except for a hint that they may be found in Kim et al 2021. (b) are these methods all inadequate as well? The background section should put more effort into discussing the inventory of existing bias correction methods, and include some indication of why they are not up to the task of correcting hydrology output here.*

We modified the text in Line 7-8:

> "… Baltic Sea. To achieve low biases in riverine freshwater inflow in large-scale climate applications, a bias correction is required that can be applied in periods where runoff observations are not available and that allows spatial transferability of its correction factors. In order to meet these requirements, we have …"

We modified the text in Line 54-55:

> "The bias correction of river runoff is an approach that has been used particularly for short term hydrological forecasts and ensemble predictions of up to six months. However, these approaches (see, e.g., those listed in Kim et al. 2021 and Madadgar et al. (2014)) are often specifically trimmed to flood forecasts. Therefore, they often require the existence of observed values from previous time steps so that that are not applicable in climate change studies, such as autoregression models (Kim et al., 2021) or components of a Bayesian forecasting system (Krzysztofowicz and Maranzano, 2004). Others like non-parametric methods based on Bayesian approaches as proposed by Brown and Seo (2010; 2012) need a large number of ensemble members (Madadgar et al. 2014).
>
> Recently, bias correction …"

We added new text in line 60:

"… (USA). A criticism of using quantile-mapping in flood forecasting is that it does not maintain the pairing of corresponding simulated and observed flows (Madadgar et al. 2014). Madadgar et al. (2014) also noted …"

We added new text after line 78:

"Therefore, we decided to not apply methods that employ detailed modifications of the discharge curves for specific rivers such as those methods that use complex matrix arithmetic of observed and simulated discharge time series (e.g. Zhao et al., 2011), or the common quantile-mapping approaches, The latter are conducted using a lot of bins, so that the bias in the discharge curve of a specific river can be strongly reduced. However, these detailed correction factors for every bin may likely not be transferred to other locations. It may work for the same river if station and river mouth are relatively close to each other, but certainly may not be valid for the transfer to neighbouring catchments."

*Line 71-73: (a) What is "a high degree of consistency"? Is it related to correlation? Is there a definition for this? (b) The proposed method applies correction factors to the simulated runoff data based on the percentile band the data lands in (discharges that get correction factor 1 notwithstanding); this is a fairly intrusive change that modifies the data significantly, as per Table 3, and requires discontinuity patching. What is the key aspect about the 3-band method that enables it to maintain a 'high degree of consistency', where other methods are deficient and would presumably degrade this consistency?*

Yes, we mean correlation. According to your remark we noticed that our wording was not very clear. Hence, we modified the text in:

Line 71:

"…high degree of temporal consistency …"

Line 73:

"…, a bias correction with as little fitting or modification of the daily sequence of runoff as possible is desired."

In addition, we added the supplementary figure S1 and the following text after line 320.

[revised manuscript text omitted]

*Lines 147-163: The article would benefit greatly from an explanation/justification about how this method differs from existing quantile mapping (QM) methods (e.g. empirical QM, delta QM, ...). If I've understood correctly, the proposed method is similar to QM: considering quintiles, the low and high bands correspond to the first and last quintile, while the middle corresponds to a merged 2[nd], 3[rd] and 4[th] quintile. Thus, the method is not that far from a quintile-based QM method without interpolation, where the middle three quintiles are merged and a workaround added to mitigate the two discontinuities. Pondering the connection: (a) if the middle three quintiles were un-merged, would it degrade the performance? (b) if linear interpolation were added, that would replace the discontinuity workaround; any reason not to do it? (c) if a and b are ok, then we've basically arrived at (a coarse) empirical quantile mapping. (d) reversing this thought process: can the present method be better described as a*

*special case of quantile mapping? E.g. tail-focused quantile mapping, or nonuniform quantile mapping, or 1:3:1 quantile mapping?*

According to your remark regarding *lines 53-66*, we added new text to clarify our objective in choosing a bias correction method, and also why we did not choose a common quantile mapping approach (please see our response above).

a) Using five percentile ranges may slightly improve the performance at the station locations but may have a degrading effect on the transfer to river mouth locations and neighbouring catchments. On purpose, we did not generate five percentile ranges as we think that a sequence of the respective five correction factor has less spatial transferability characteristics than the three we are using. As the spatial transfer of bias correction factors for river runoff is a rather new approach (see our response to your comment on line 174-180 below), we cannot undermine this statement by other literature. However, our evaluation of inflow into sea basins is supporting the validity of our approach.

b) The discontinuity work around is actually a linear interpolation of the correction factors between the two neighbouring percentile ranges (low & middle, middle & high).

c) and d) In a wider sense, the three-part bias correction corresponds to a very coarse quantile-mapping. However, designating it as quantile mapping approach may suggest that a detailed mapping is done as in the commonly used quantile-mapping applications. One the one hand, we want to avoid a potential mislead of the reader. On the other hand, we see your point on the relationship with quantile mapping approaches. Consequently, we renamed our method and modified the title to pay regard to your remark:

New title**: A three-quantile bias correction with spatial transfer for the correction of simulated European river runoff to force ocean models**

Within the text, we will mainly replace 'three-part' by 'three-quantile' bias correction.

We are also pointing to the correspondence with a coarse quantile mapping approach in the beginning of Sect. 2.4. Here, we modified the text in lines 147-148:

"We have developed a bias correction method for river runoff that uses correction factors for three quantiles and includes a spatial transfer of these factors. We note that our three-quantile bias correction is similar to a very coarse quantile mapping. The latter has been introduced in climate change impact research to correct for significant biases in data produced by global and regional climate models. Quantile mapping is a distribution mapping in which the distribution function of climate values is corrected to match the observed distribution function. Details of such mapping applied to precipitation and surface air temperature can be found, for example, Piani et al. (2010) and Teutschbein et al. (2011). Our bias correction method involves several steps. First, different correction factors for low, medium and high percentiles are calculated at the station locations and then applied at the respective river mouths."

*Line 174-180: The method of interpolating bias correction factors from gauged river mouths to not-gauged river mouths, up to a maximum distance, seems reasonable. Is this an established approach, or is it part of the present method? Are there any references to what other hydrologists have done for transferring bias corrections to ungauged rivers?*

We added the following text after the text inserted in line 78 in response to your comment on line 53-66:

> "We note that the spatial transfer of bias correction factors for river runoff on a large scale is a rather new approach. Within a specific river, Lakew and Moges (2021) applied spatial interpolation of bias correction factors within the Upper Blue Nile system based on 12 gauging stations. Nijssen et al. (2020) trained a machine learning (ML) model to perform site-specific bias corrections in the Columbia River and then applied the ML model to river reaches without flow observations."

*Line 267-269: What was used for the initial conditions -- presumably ORAS5?*

ORAS5 is used as the lateral boundary conditions. The initial conditions were taken from a 20-years spin-up simulation driven by ERA5 data. We added the following text to line 269:

> "… 2018. Initial conditions were taken from a 20-years spin-up simulation driven by ERA5 data, so that the deeper ocean layers could adapt to the present-day climate (S. Grayek, pers. comm., 2023). Note that for the evaluation of results, we neglected the year 2010 to have an additional spin-up where NEMO could adapt to the specific transient conditions within each of the two experiments. For the German Bight, this spin-up of one year is sufficient as the residence time of water may comprise only up to four months (Becker et al., 1999). In the two experiments, the …"

*Line 325-327: on 'selected rivers', there is no rationale given for the river selection. There must be a reason those ones were chosen*

We modified the text:

> "… discharge during 1979-2014 for selected rivers, where the three-quantile bias correction led to a noticeable KGE improvement in comparison to the mean bias correction."

*Line 487-488: The simulation starts in 2010, and the evaluation also starts in 2010. There should be a "spin up" time scale here where you allow the model to adjust to the new forcing and evaluate after a number of those time scales have elapsed (eg, 2-3). What is the residence time for surface waters in the German Bight area? This time scale may be fast (days? weeks? months?) but should be included for context. Meanwhile, same question for the deeper waters – is the residence time here large or small? This is for putting "not much happened below 30m" into context.*

We provided information on the ocean model initial conditions in response to your comment on 267-269. Here, we also state that we now used the year 2010 as an additional spin-up so that our evaluation comprises the period 2011-2018. We updated our figures and the text accordingly. Surface waters in the German Bight are exchanged within a few days. For deeper layers, the residence time of water may comprise up to four months (Becker et al. 1999). The latter is now also referred in the new text mentioned above.

In addition, we added text in line 510:

> "… (not shown). This is consistent with the fact that the *Deutsche Bucht* and *EMS* stations are located in an area where the salinity is temporarily stratified, depending on

the meteorological conditions and the intensity of river runoff (Klein and Frohse, 2008)."

*Line 495-496: Evaluating mean sea-surface salinity (SSS) over eight years will wipe out the variability. I would expect the 8-year mean SSS bias to be impacted by biases in the mean freshwater input, and less so by the variability in the freshwater input. As in the general comments section, this particular evaluation does not showcase well the 3-band bias correction method's strengths. If you run the ocean simulation with simple bias correction (e.g. Table 3 middle part), or with the climatology freshwater datasets, you should get the 8-year mean SSS comparable to that from the 3-band method. If I've got this wrong, then showing the climatology or simple bias correction cases alongside would strengthen the case for the 3-band method. Meanwhile, as I understand it the strength of the 3-band method is the better capturing of the variability -- and indeed Figures 7,8,10 support this -- so one should look for better salinity /variability/ in the ocean model simulations: (a) are there runoff events or interannual variability in discharge (e.g. Fig7 top panel) that appear, perhaps with some lag, as salinity drops in the DB and EMS station data, and are these signals better represented with the bias-corrected ocean model run? (b) similar question but for the satellite-based surface salinity product; are runoff events / interannual variability better represented when looking at shorter time scales than 8 yr mean – daily, weekly, seasonally, interannually?*

We added the new Figure 15 and the supplementary Figure S2, and we added text in line 501:

"Similar improvements can also be seen in June 2013 when the Elbe flood is strongly influences the SSS of the German Bight (Figure S2). Here, the increase in salinity due to the bias corrected runoff (Figure S2b) is more pronounced than in the long-term mean (Figure 13b). In addition, we found that use of the bias corrected river runoff also improves the SSS variability expressed by its coefficient of variation, shown in the *new* Figure 15."

[Figure]

**Figure S2**. Same as Figure 13, but with SSS averages calculated over the period June 2013.

[Figure]

new **Figure 15**: Coefficients of variation of SSS in the German Bight for the period from 2011-2018: a) OBS, b) HDW and c) HD-BC

*Line 507-508: If this is actually RMSE then it is unclear if the reduction in RMSE is due to reduced error in variability or due to reduced bias: RMSE^2 = CRMSE^2 + bias^2. Additionally reporting CRMSE here (and/or another mean-removing metric such as correlation, or gamma^2), would better capture improvement in variability.*

We added the new table 6 and the new table S1 in the supplement, and we modified the text in line 507-508:

[revised manuscript text omitted]

*Lines 518-530: This reads like the 3.6 km model is too coarse to get good nearshore salinity and is not up to the task. Are there any higher-resolution NEMO models available for the German Bight that could be used as a higher-resolution downscale? That is, to capture some of the estuaries and better resolve the coastline*

We choose this setup as this is a standard setup of NEMO used over this region (see, e.g., Bonaduce et al., 2020; Grayek et al., 2023; Ho-Hagemann et al., 2020; Nguyen et al., 2024) at our institute. In principle, it is possible to conduct simulations with NEMO on a higher resolution e.g. 1 km covering only the German Bight or at 2.5 km covering the North Sea. However, currently we do not have any experience with such a setup. For the latter, there is an on-going cooperation with the Federal Maritime and Hydrographic Agency where such a simulation is planned for the year 2018 (M. Ricker, pers. communication, 2024). Therefore, conducting such a simulation is beyond the scope of the present study and may be done in the future. In addition, a major motivation of developing the bias correction was its application within our regional coupled system model GCOAST-AHOI (Ho-Hagemann et al., 2020), such as mentioned in line 600-601.

*Lines 587-590: This should be reworded to be more precise about what aspects of the sea-surface salinity were improved (e.g., the 8 yr mean, and pending resolution of comments for line 507-508, variability at stations)*

We added the following text in line 590:

"…coastal areas, especially regarding the mean salinity and its variability."

*Lines 593-603: the 3-band method is similar to quantile mapping (as per comments above for lines 147-163), and quantile mapping has been flagged as potentially not suitable for climate simulations as it has been shown to degrade trends (i.e. references in Cannon 2015), where Cannon 2015 proposed a delta QM that preserves trends (by extracting them, applying the quantile mapping, and reinserting them). The 3-band method proposed here does not take any special care about preserving trends. Adding some rationale for why this method is applicable to climate simulations, particularly when contrasted with other bias correction methods that are not applicable, would strengthen this conclusion.*

Maraun et al. (2017) pointed out that a debate has arisen about whether trend modification by variance-adjusting bias correction methods actually improves or degrades the raw climate change signal. They further argued that purely statistical arguments cannot resolve this issue, which requires process understanding. With respect to runoff, the latter needs to take into account the spatial and temporal characteristics of rivers and events, which is beyond the scope of the present large-scale study. However, we have added a new section 3.4 to address this debate:

"3.4 Effect of the bias correction on contemporary trends

As mentioned in Sect. 2.4, our three-quantile bias correction is similar to a very coarse quantile mapping, and quantile mapping has been flagged as potentially not suitable for climate simulations as it has been shown to modify trends (e.g. references in (e.g. references in Cannon et al., 2015). However, Maraun et al. (2017) pointed out that a debate has arisen about whether trend modification by variance-adjusting bias correction methods actually improves or degrades the raw climate change signal. They further argued that purely statistical arguments cannot resolve this issue, which requires process understanding. With respect to runoff, the latter needs to take into account spatial and temporal characteristics of rivers and events, which is beyond the scope of the present large-scale study.

To investigate the effect of the bias correction on contemporary trends, we calculated trends in the annual maximum, mean and minimum discharge for the period 1979-2014 and compared the results for HDW and HDW-BC (new Fig. 10). The trend patterns are generally within the range spanned by the two datasets considered in Hagemann and Stacke (2022). For the annual maximum and mean discharge, the trend patterns are only slightly changed by the bias correction. For the annual minimum discharge, the trend pattern is quite similar in HDW and HDW-BC. However, there are a few more rivers where the magnitude of the trend is affected by the bias correction. This is particularly the case over Scandinavia where many rivers are regulated, so that that the correction of the low percentile range is often strong to account for the effect of regulation on low flows (cf. Sect. 3.2)."

[Figure]

New **Figure 10**: Trends in annual maximum (1st row), mean (2nd row) and minimum (3rd row) discharge [%/a] for HDW (left column) and HDW-BC (right column) from 1979-2014.

**Technical corrections**

*Line 26: include ocean keywords such as nearshore or sea-surface salinity, or ocean model?*

We added sea-surface salinity.

*Line 134, the reference is to a Zenodo link to the HD model code. This seems more appropriate for the data/code availability section, and the early part of sec 2.3 should explain the HD model*

In Line 134, we cite Hagemann et al. (2020), which is the first scientific publication of the high-resolution HD model published in Front. Earth Sci. The HD code publication in Zenodo is

Hagemann et al. (2023). We assume that the reviewer accidentally looked at the wrong reference.

*Line 119: "0.5 spatial resolution" requires units*

We added the missing unit:

"… 0.5° spatial resolution …"

*Line 281: KGE defined here, no need to de-acronym it later (lines 308, 331, 381, 405)*

We replaced the full name by its abbreviation in the mentioned places.

*Line 417: "Observed and simulated daily discharge of HD5-GSWP3" – surely this should be discharge of water*

Yes. We modified the caption to avoid confusion:

"Observed and simulated daily discharge based on HD5-GSWP3 …"

*Line 510-512: if the NSB station is not used/usable in the eval, remove it entirely*

We removed the NSB station from the updated Figure 13 (see below) and the respective text in lines 505 and 510-512.

[Figure]

**Figure 13** Mean analyzed SSS: a) Observations (OBS) and various SSS differences of the NEMO experiments in the German Bight for the period from 2011 to 2018. The SSS differences comprise b) HD-BC minus HDW, c) HDW minus OBS, and d) HD-BC minus OBS. The land boundaries are displayed by the dashed grey lines.

*Sec 2.3: Can the HD5-WFDE5 and HD5-GSWP3 acronyms be shortened? H5 and H3 perhaps?*

We replaced HD5-WFDE5 by HDW, HD5-GSWP3 by HDG, and HD5-Bias C. by HD-BC. In this respect, HDW-BC and HDG-BC refer to the bias corrected data of HDW and HDG, respectively.

*Figure 7,8,10: use (a), (b), etc for the panels instead of first panel, second panel*

Corrected as suggested.

*Figures, general: in the copy of the manuscript received for review, the figures appear to be JPEGs. Suggest to switch to a vector format or PNG*

For the submission of the revised version, we will take care that the figure quality/resolution is appropriate.

**Added references**

[revised manuscript text omitted]

---

## Author Comment (AC2)

**Reply to the comments of reviewer 2**

We thank the reviewer for the thorough reading of the manuscript and the valuable remarks that helped us to improve the manuscript. In the following, the original reviewer comments are given in italic and all line numbers and figure numbers refer to the original submitted version that was reviewed if not mentioned otherwise.

**Reply to review of reviewer 2**

*Detailed river runoff data are needed for the coastal ocean models. Unfortunately, the observational data are often with too coarse resolution and/or of insufficient accuracy. Data coverage may be improved by hydrological modelling, but it creates specific errors. As a result of freshwater input bias, modelled ocean state may drift away from the real salinities and related variables, especially in climate-related long-term studies. Therefore, the approach of the present MS to elaborate and test bias correction for simulated river runoff is highly needed and scientifically interesting.*

*Obtained results are convincing. In particular, the model system gives too large peak discharges in the Elbe, Rhine, Weser and Odra rivers, while the modelled low discharges are close to the observed values. This range-dependent bias is effectively corrected by the applied method.*

*The MS has a good quality and should be published. When reading I found some problems, resolving of which in the revised MS would help the readers.*

Thank you for the positive evaluation of our manuscript.

A) *One problem lies within the title, that brings forward "three-part". It is not clear what does it tell scientifically; what is the difference from "two-part" or "four-part" bias correction? I would prefer a scientific name for the developed bias corrected method. Is it the "quantile mapping bias correction"? Later it has been explained that the method has been specifically applied to "different correction factors for low, medium and high percentile ranges of river runoff over Europe". Note that in the literature there are numerous uses of "three-step" bias correction that differ from the basic principles as well as from applications. Another possible interpretation of "three-part" is: (1) quantile mapping and correction at measurement sites, (2) transfer of bias correction to the river mouths, (3) interpolation of bias to the unsampled coastal sections. In short, the MS would benefit from having a well-defined title, not leaving space for ambiguity.*

We modified the title to pay regard to your concerns:

New title**: A three-quantile bias correction with spatial transfer for the correction of simulated European river runoff to force ocean models**

Within the text, we will mainly replace 'three-part' by 'three-quantile' bias correction.

B) *Quantile mapping is a well-known approach for bias correction in climate studies. The MS would benefit, if some basic statistical concepts and references are introduced in the introduction. Presently, the introductory paragraph in lines 53-66 is rather fragmentary and does not provide sufficient background information for the study. Specific to the river discharge, this paragraph is too much listing specific case studies and is lacking the references to the well-known studies, except for Madadgar et al (2014).*

We modified several parts of our introduction to make the aim of our study clearer and why we chose the bias correction method that is presented here.

We modified the text in Line 7-8:

"… Baltic Sea. To achieve low biases in riverine freshwater inflow in large-scale climate applications, a bias correction is required that can be applied in periods where runoff observations are not available and that allows spatial transferability of its correction factors. In order to meet these requirements, we have …"

We modified the text in Line 54-55:

"The bias correction of river runoff is an approach that has been used particularly for short term hydrological forecasts and ensemble predictions of up to six months. However, these approaches (see, e.g., those listed in Kim et al. 2021 and Madadgar et al. (2014)) are often specifically trimmed to flood forecasts. Therefore, they often require the existence of observed values from previous time steps so that that are not applicable in climate change studies, such as autoregression models (Kim et al., 2021) or components of a Bayesian forecasting system (Krzysztofowicz and Maranzano, 2004). Others like non-parametric methods based on Bayesian approaches as proposed by Brown and Seo (2010; 2012) need a large number of ensemble members (Madadgar et al. 2014).

Recently, bias correction …"

We added new text in line 60:

"… (USA). A criticism of using quantile-mapping in flood forecasting is that it does not maintain the pairing of corresponding simulated and observed flows (Madadgar et al. 2014). Madadgar et al. (2014) also noted …"

We added new text after line 78:

"Therefore, we decided to not apply methods that employ detailed modifications of the discharge curves for specific rivers such as those methods that use complex matrix arithmetic of observed and simulated discharge time series (e.g. Zhao et al., 2011), or the common quantile-mapping approaches, The latter are conducted using a lot of bins, so that the bias in the discharge curve of a specific river can be strongly reduced. However, these detailed correction factors for every bin may likely not be transferred to other locations. It may work for the same river if station and river mouth are relatively close to each other, but certainly may not be valid for the transfer to neighbouring catchments."

C) *Introductory paragraph in lines 37-52 could be made more informative, backing the statements with appropriate references. In particular, the principle that "the river runoff is consistent with the atmospheric forcing" could be more elaborated and referenced. How bias correction changes this consistency? (Later this question is briefly discussed in lines 71-78).*

We modified the text in line 41:

"… atmospheric forcing (e.g. Vinayachandran et al., 2015; Hagemann and Stacke, 2022), i.e. that the impact of weather events and trends in the atmospheric forcing is transferred via the river runoff into the ocean."

In order to clarify that we specifically consider temporal consistency, we modified the text and added some analysis on how our bias correction affects this consistency. Hence, we modified the text in:

Line 71: "…high degree of temporal consistency …"

Line 73: "…, a bias correction with as little fitting or modification of the daily sequence of runoff as possible is desired."

In addition, we added the supplementary figure S1 and the following text after line 320.

"In order to analyse how much the bias correction affects the daily sequence of river runoff at the station locations, we calculated the correlation between the simulated discharges and the observations. Supplementary Figure S1 shows that the correlation patterns of HDW and HDW-BC with observed discharges are quite similar. For rivers where differences can be identified, the correlation mostly increases for HDW-BC. The correlation between HDW and HDW-BC is generally higher than 0.95, and only a very few rivers show correlations lower than 0.9. These rivers are usually rivers that are heavily influenced by human activities, such as the Volga and the Luleaelven."

We also modified the text in line 281-282:

"… mean bias, the Pearson correlation coefficient and the Kling-Gupta efficiency (KGE; Gupta et al., 2009; Kling et al., 2012). All metrics …"

D) *The sub-section "2.4 Bias correction of river runoff" lacks references to the basic principles. It should have clear description of already known procedures (with references) and new/specific approaches. If the method is completely novel, this could be spelled out; then also more justification should be presented.*

We modified the text in lines 147-148:

"We have developed a bias correction method for river runoff that uses correction factors for three quantiles and includes a spatial transfer of these factors. We note that our three-quantile bias correction is similar to a very coarse quantile mapping. The latter has been introduced in climate change impact research to correct for significant biases

in data produced by global and regional climate models. Quantile mapping is a distribution mapping in which the distribution function of climate values is corrected to match the observed distribution function. Details of such mapping applied to precipitation and surface air temperature can be found, for example, Piani et al. (2010) and Teutschbein et al. (2011). Our bias correction method involves several steps. First, different correction factors for low, medium and high percentiles are calculated at the station locations and then applied at the respective river mouths."

E) *Lines 181-183 say that "bias correction can lead to spurious daily jumps in discharge when the percentile boundary is crossed and the bias correction factors differ between the percentile ranges". This is suppressed by applied smoothing. It would be interesting to know if these jumps could be avoided (not suppressed) by some elaboration of the methods, for example by introducing continuous correction factors instead of stepwise correction.*

Using continuous correction factors might improve the performance of the bias correction at the station locations but may have a degrading effect on the transfer to river mouth locations and neighboring catchments. On purpose, we did not generate more than three percentile ranges as we think that the higher the number of correction factors is, the less spatially transferable they are. As the spatial transfer of bias correction factors for river runoff is a rather new approach (This, we point our more clearly in our response to the comment of reviewer 1 on line 174-180.), we cannot undermine this statement by other literature. However, our evaluation of inflow into sea basins is supporting the validity of our approach.

Note also that the smoothing work around is actually a linear interpolation of the correction factors between the two neighboring percentile ranges (low & middle, middle & high).

**Some technical remarks.**

*1. Principles and approaches of the HydroPy model and HD model (is it a unique name?) should be shortly outlined in the beginning of sections 2.2 and 2.3.*

Yes, HD model is a unique name that has not been duplicated since the publication of its first version by Hagemann and Dumenil (1998).

We added in line 121:

"HydroPy (Stacke and Hagemann, 2021) is a state-of-the-art global hydrology model for which no model calibration was performed for its setup. Within global hydrological modelling, the usage of uncalibrated models is rather common (see, e.g., Haddeland et al., 2011), even though some models exist that are calibrated for global studies. In the present study, HydroPy was driven …"

We added in line 134:

"The HD model (Hagemann et al., 2020) is a well-established river routing model that is implemented in a range of global and regional model systems. As noted in Hagemann et al. (2020), no river specific parameter adjustments were conducted in the HD model

to enable its applicability for climate change studies and over catchments, where no daily discharges are available at a downstream station. To simulate discharge with the HD model, we used …"

Please note that we originally planned to add more details in the sections 2.2 and 2.3. However, the editor correctly pointed out that these details comprised a lot of doubling of text with the information provided in Hagemann et al. (2020) and Hagemann and Stacke (2022). Consequently, we removed these details and referred to the respective publications.

*2. Smoothing formulae in lines 185-188 have different notations for q and Q than in lines 153-155.*

In lines 185-188, we corrected the notation by using $Q_p$ and $Q_{100-p}$.

*3. Titles of Table 5 and Figure 12 contain "fractional area coverage" that is not defined.*

We modified the titles of Table 5 and the caption of Figure 12:

"… fractional catchment coverage (see Table 1) is …"

*4. Legends of Figures 5, 7-10 contain "HD5.2-HydroPy-WFDE5" that are not defined.*

According to the comments of reviewer 1, we replaced HD5-WFDE5 by HDW, HD5-GSWP3 by HDG, and HD5-Bias C. by HD-BC. In this respect, HDW-BC and HDG-BC refer to the bias corrected data of HDW and HDG, respectively. In the respective figures, HD5.2-HydroPy-WFDE5 is replaced by HDW and HD5.2-HydroPy-WFDE5 by HDG.

*5. I counted 37 abbreviations; alphabetically from "20CR" to "WFDE5". Some of the abbreviations are well-known like ECMWF, ESA, HELCOM, NASA, OSPAR and they do not complicate the reading. At the same time, there are abbreviations that occur only once (AHOI, GCOAST, RCSM, they appear only in the conclusions, BSH, RSME – they are not defined, GSM) or a few times (DB, EMS, GRDC, ISIMIP ...). Reader would benefit from less abbreviations.*

We reduced the number of abbreviations:

We removed 20CR, GCOAST, AHOI, ESA; GRDC, GSM, NASA; RCSM, SMAP; SMOS; WFD, BSH, DKRZ, WDCC and used only their respective full names. We will check the manuscript for further obsolete abbreviations.

We added the definition of RSME in Sect. 2.7

**Added references**

Brown, J. D., and Seo, D. J.: A nonparametric postprocessor for bias correction of hydrometeorological and hydrologic ensemble forecasts, J. Hydrometeorol., 11, 642-665, https://doi.org/10.1175/2009jhm1188.1, 2010.
Brown, J. D., and Seo, D. J.: Evaluation of a nonparametric post-processor for bias correction and uncertainty estimation of hydrologic predictions, Hydrol. Process., 27, 83-105, https://doi.org/10.1002/hyp.9263, 2012.

Haddeland, I., Clark, D. B., Franssen, W., Ludwig, F., Voß, F., Arnell, N. W., Bertrand, N., Best, M., Folwell, S., Gerten, D., Gomes, S., Gosling, S. N., Hagemann, S., Hanasaki, N., Harding, R., Heinke, J., Kabat, P., Koirala, S., Oki, T., Polcher, J., Stacke, T., Viterbo, P., Weedon, G. P., and Yeh, P.: Multimodel estimate of the global terrestrial water balance: setup and first results, J. Hydrometeorol., 12, 869-884, https://doi.org/10.1175/2011jhm1324.1, 2011.

Krzysztofowicz, R., and Maranzano, C. J.: Hydrologic uncertainty processor for probabilistic stage transition forecasting, J Hydrol, 293, 57-73, https://doi.org/10.1016/j.jhydrol.2004.01.003, 2004.

Piani, C., Weedon, G. P., Best, M., Gomes, S. M., Viterbo, P., Hagemann, S., and Haerter, J. O.: Statistical bias correction of global simulated daily precipitation and temperature for the application of hydrological models, J Hydrol, 395, 199-215, https://doi.org/10.1016/j.jhydrol.2010.10.024, 2010.

Taylor, K. E.: Summarizing multiple aspects of model performance in a single diagram., J Geophys Res-Atmos, 106, 7183-7192, https://doi.org/Doi 10.1029/2000jd900719, 2001.

Teutschbein, C., Wetterhall, F., and Seibert, J.: Evaluation of different downscaling techniques for hydrological climate-change impact studies at the catchment scale, Clim. Dyn., 37, 2087-2105, https://doi.org/10.1007/s00382-010-0979-8, 2011.

Vinayachandran, P. N., Jahfer, S., and Nanjundiah, R. S.: Impact of river runoff into the ocean on Indian summer monsoon, Environmental Research Letters, 10, https://doi.org/10.1088/1748-9326/10/5/054008, 2015.

Zhao, L., Duan, Q., Schaake, J., Ye, A., and Xia, J.: A hydrologic post-processor for ensemble streamflow predictions, Advances in Geosciences, 29, 51-59, https://doi.org/10.5194/adgeo-29-51-2011, 2011.

---

## Author Comment (AC3)

[Figure]

**Figure S1**: Correlation of a) HDW and b) HDW-BC with observations as well as c) HDW with HDW-BC from 1979-2014.

---

## Author Response (AR2)

**Reply to the comments of both reviewers**

We thank the reviewers for their positive evaluation of the revised version of our manuscript.

**Reply to review of reviewer 1 (report 2)**

We thank the reviewer for his remark. In the following, the original reviewer comments are given in italic and all line numbers are provided for the revised version that was reviewed, and for the new updated version in brackets.

*- Spatial transferability of correction factors to nearby watersheds is a key part of the posed method, and I agree with the authors that directly transferring fine-grained quantile mapping based coefficients is unlikely to be a satisfactory approach (lines 98-101 in revised manuscript). The soft language of "may likely not" and "may not be valid" is fine for discounting the fine-grained QM transferability, but the article lacks stronger language supporting three-quantile transferability. (A) The bulk of the support seems to be lines 221-227 which refers to Section 3.1 (presumably for lines 363-372), wherein it sounds like the three-quantile method is correcting for biases in the atmospheric forcing dataset, where these biases are order 200 km in spatial scale, giving rise to the 200 km transferability radius. Is it the correct intuition backing transferability? Suggest to make this connection in the text if so. (B) If not, is there any other way to strengthen the transferability case? For ex, is there an experiment where you spatially transfer correction factors to /gauged/ rivers, such that you could compare gauged river data with the bias-corrected data and the spatially-transferred bias-corrected data?*

We added the following text in line 373 (372):

We also note the large-scale patterns of positive and negative discharge biases (Figure 5). Abrupt changes in bias behaviour along the same coastline are rare. Most of the few cases can be attributed to large human water abstractions from the river, i.e. especially for the Ebro River (see also Section 3.3) and in Turkey, which are not considered by the model. This supports our assumption about the spatial transferability of the three-quantile bias correction factors. The bias patterns are related to biases in the atmospheric forcing dataset or biases introduced by the HydroPy model.

In addition, we added the following sentence in line 689 (694) in the Summary and Conclusions Section 5.

... areas. The improved inflows to the sea basins, together with the fact that the discharge bias behaviour tends not to vary abruptly along the same coastline, underpin the validity of our transferability approach. Exceptions ...

*- Repeated text in line 476*

Repeated text in line 376 (381) has been removed.